# Reprogramming of the antimycin NRPS-PKS assembly lines inspired by gene evolution

Takayoshi Awakawa[1,2], Takuma Fujioka[1], Lihan Zhang[1], Shotaro Hoshino[1], Zhijuan Hu[1], Junko Hashimoto[3], Ikuko Kozone[3], Haruo Ikeda[4], Kazuo Shin-Ya[2,5], Wen Liu[6] & Ikuro Abe [1,2]

Reprogramming of the NRPS/PKS assembly line is an attractive method for the production of new bioactive molecules. However, it is usually hampered by the loss of intimate domain/module interactions required for the precise control of chain transfer and elongation reactions. In this study, we first establish heterologous expression systems of the unique antimycin-type cyclic depsipeptides: JBIR-06 (tri-lactone) and neoantimycin (tetra-lactone), and engineer their biosyntheses by taking advantage of bioinformatic analyses and evolutionary insights. As a result, we successfully accomplish three manipulations: (i) ring contraction of neoantimycin (from tetra-lactone to tri-lactone), (ii) ring expansion of JBIR-06 (from tri-lactone to tetra-lactone), and (iii) alkyl chain diversification of JBIR-06 by the incorporation of various alkylmalonyl-CoA extender units, to generate a set of unnatural derivatives in practical yields. This study presents a useful strategy for engineering NRPS-PKS module enzymes, based on nature's diversification of the domain and module organizations.

[1] Graduate School of Pharmaceutical Sciences, The University of Tokyo, 7-3-1 Hongo, Bunkyo-ku, Tokyo 113-0033, Japan. [2] Collaborative Research Institute for Innovative Microbiology, The University of Tokyo, Yayoi 1-1-1, Bunkyo-ku, Tokyo 113-8657, Japan. [3] Japan Biological Informatics Consortium, 2-4-7 Aomi, Koto-ku, Tokyo 135-8073, Japan. [4] Laboratory of Microbial Engineering, Kitasato Institute for Life Sciences, Kitasato University, Kanagawa 252-0373, Japan. [5] National Institute of Advanced Industrial Science and Technology (AIST), 2-4-7 Aomi, Koto-ku, Tokyo 135 0064, Japan. [6] State Key Laboratory of Bio-Organic & Natural Products Chemistry, Shanghai Institute of Organic Chemistry, Chinese Academy of Sciences, Lingling Road 345, 200032 Shanghai, China. Correspondence and requests for materials should be addressed to T.A. (email: awakawa@mol.f.u-tokyo.ac.jp) or to K.S-Y. (email: k-shinya@aist.go.jp) or to I.A. (email: abei@mol.f.u-tokyo.ac.jp)

Polyketides and non-ribosomal peptide hybrids are an important class of natural products, and include a variety of bioactive compounds, exemplified by the anti-cancer agents, bleomycin, epothilone, and calyculin[1–4]. They are built up by a complex system composed of polyketide synthases (PKSs) and non-ribosomal peptide synthetases (NRPSs)[5–7], which are thiotemplate module enzymes consisting of several catalytic domains. In the PKS system, the ketosynthase (KS) domain catalyzes decarboxylative Claisen condensation between a starter acyl group and an extender unit thioesterified to an acyl carrier protein (ACP) via the action of an acyltransferase (AT), during a single polyketide extension cycle. In the NRPS system, the amino acid building blocks are activated as adenylates and thioesterified to the thiolation (T) domain by the adenylation (A) domain. The condensation (C) domain forms a peptide bond between two acylated T domains during a single peptide extension cycle. In both systems, if the termination domain is a thioesterase (TE), then chain release typically occurs by hydrolysis or macrocyclization. A set of domain complexes responsible for one cycle of chain extension is called a module. The amino acid sequences within subunits that join domains and modules covalently are referred to as a linker, and those which are located on distinct subunits and mediate communication between modules are referred to as a docking domain[8,9]. In the NRPS–PKS complex systems, the product of the NRPS module is transferred onto the KS domain of PKS module and condensed with a polyketide extender unit by PKS, or the product of PKS module is condensed with an amino acid to form a peptide bond by NRPS. The substrate specificities of the KS and C domains are usually rigid, thus contributing to the maintenance of the precise chain transfer order. Several reports have clarified the reaction mechanism of each module and domain in the NRPS and PKS systems in terms of the protein structure[6]; however, the engineering of these systems usually accompanies significant loss of productivity due to the strict regulation of the module enzymes[10].

To overcome such problems, the optimal joint connections for each catalytic domain and module should be identified. Recently, Bode and co-workers reported a new strategy for the manipulation of NRPSs by using exchange units that are sets of A-T-C domains that can be transplanted into active chimeric modules[11]. However, this strategy may not be necessarily applied to hybrid NRPS-PKS systems because the quaternary organization of NRPS-PKS is likely different from that of pure NRPS system. In fact, the quaternary structure of NRPS module is defined as a monomer in the early chromatographic[12] and X-ray crystallographic studies[13], but we cannot exclude that it works as a dimer in the NRPS-PKS system, as the docking domain from tubulysin system forms a homodimer in the solution[9]. It has thus remained challenging to establish a general rule for rational reconstruction of module assembly lines, and scientists still rely on trial-and-error strategies. One possible solution is to learn how nature diversifies the domain and module organizations by taking advantage of bioinformatics analyses, through comparisons of closely related enzymes. In nature, various module enzymes have evolved through horizontal gene transfer, gene deletion, mutation, and so on[14–17] to form functional sets of modules. Thus, for cis-AT PKSs, the predominant evolutionary model is a repeated cycle of duplication of ancestral modules coupled with domain exchanges[14,15,17], while for the distinct class of trans-AT PKSs, it is rather horizontal gene transfer that appears to dominate[16,17]. However, the specific evolutionary history of hybrid cis-AT PKS/NRPS system has yet to be rigorously investigated. It is a promising way to modify the module compositions according to the evolutionary course, because this method is effective to maintain the connectivity between multiple modules while minimizing the change of the overall structure and the protein–protein interactions which underlie their function. The knowledge on the constraint for NRPS-PKS engineering should be accumulated more to understand how this system has been evolved and can be manipulated. This could lead to computational platforms which clearly predict domain/linker/docking-domain boundaries as the one established in cis-PKS system for designing chimeric modules[18].

Antimycins (1) are a group of di-lactone depsipeptides with various important biological properties, including anti-fungal, insecticidal, anti-cancer, and anti-inflammatory activities[19]. They are biosynthesized by the NRPS-PKS system from substrates including a unique 3-formamidosalicylate (3-FSA) starter unit, L-threonine, pyruvate, and alkylmalonyl-CoA (Fig. 1). Their structures are characterized by the depsipeptides constructed by two NRPS modules (AntC), including one module with an unusual ketoreductase (KR)-domain, one PKS (AntD) that generates alkyl group variations at C-9 by the incorporation of a broad range of extender substrates provided by a crotonyl-CoA reductase/carboxylase (CCR)[20–23] (AntE), and a reductase (AntM). Zhang's group and we have created ~100 antimycin derivatives, by employing the biosynthetic enzymes with the tolerant substrate specificities in the antimycin systems[23–27]. Interestingly, in nature, there are tri-lactone, tetra-lactone, and penta-lactone antimycin-type depsipeptides that also utilize 3-FSA as a common starter unit[19], including the tri-lactone JBIR-06 (2)[28] and the tetra-lactone neoantimycin A (3a)[29] (Fig. 1), known downregulators of GRP78, a molecular chaperone related to resistance to chemotherapy and hypoglycemic stress[30,31]. In 2013, Magarvey's group reported two NRPS-PKS gene clusters based on bioinformatic analyses[32], and proposed that they are involved in the biosyntheses of 2 and 3a, respectively; however, their proposal has not been validated experimentally and the sequence information has not been deposited in GenBank. The NRPS-PKS assembly line organizations involved in the biosyntheses of 1, 2, and 3a are very similar, and the conservation of the two modules that incorporate 3-FSA and L-threonine in these biosynthetic systems implies that they are generated from the same ancestor. Very recently, Zhang's group also reported the neoantimycin gene cluster and showed the extender substrate of its PKS module was malonyl-CoA[33]. Here, we control the size of lactone rings of JBIR-06 and neoantimycin, and introduce diverse alkyl groups into the JBIR-06 scaffold, by rationally modifying their assembly lines with the aid of the information how they are diversified, to yield various bioactive antimycin-type depsipeptides.

## Results

**JBIR-06 and neoantimycin biosynthetic gene clusters**. In this work, we independently identified the biosynthetic gene clusters of the tri-lactone JBIR-06 and the tetra-lactone neoantimycin (the *sml* and *nat* clusters) from each producing strain (Supplementary Figs. 1–2), and heterologously expressed them in *Streptomyces lividans* TK21 to identify their products. The JBIR-06 and neoantimycin gene clusters were each cloned into pKU518[34,35], a bacteriophage-based BAC vector, resulting in pKU518J06 and pKU518nant, respectively (Supplementary Note 1, Supplementary Tables 1–2). From the sequence data of these two vectors, we confirmed that the assembly line organizations of JBIR-06 and neoantimycin NRPS-PKSs were identical with those in the previous report[32], except for NatD (Fig. 1). In contrast to the Magarvey's report, where *natD* and *natE* encode A-KR and T-TE domains, respectively, *natD* encodes a whole single module consisted of C-A-KR-T-TE domains in our sequence data. As Zhang's group also reported the same NatD domain organization as ours[33], the current module organization of neoantimycin is more plausible. HPLC analyses of the culture extracts of the

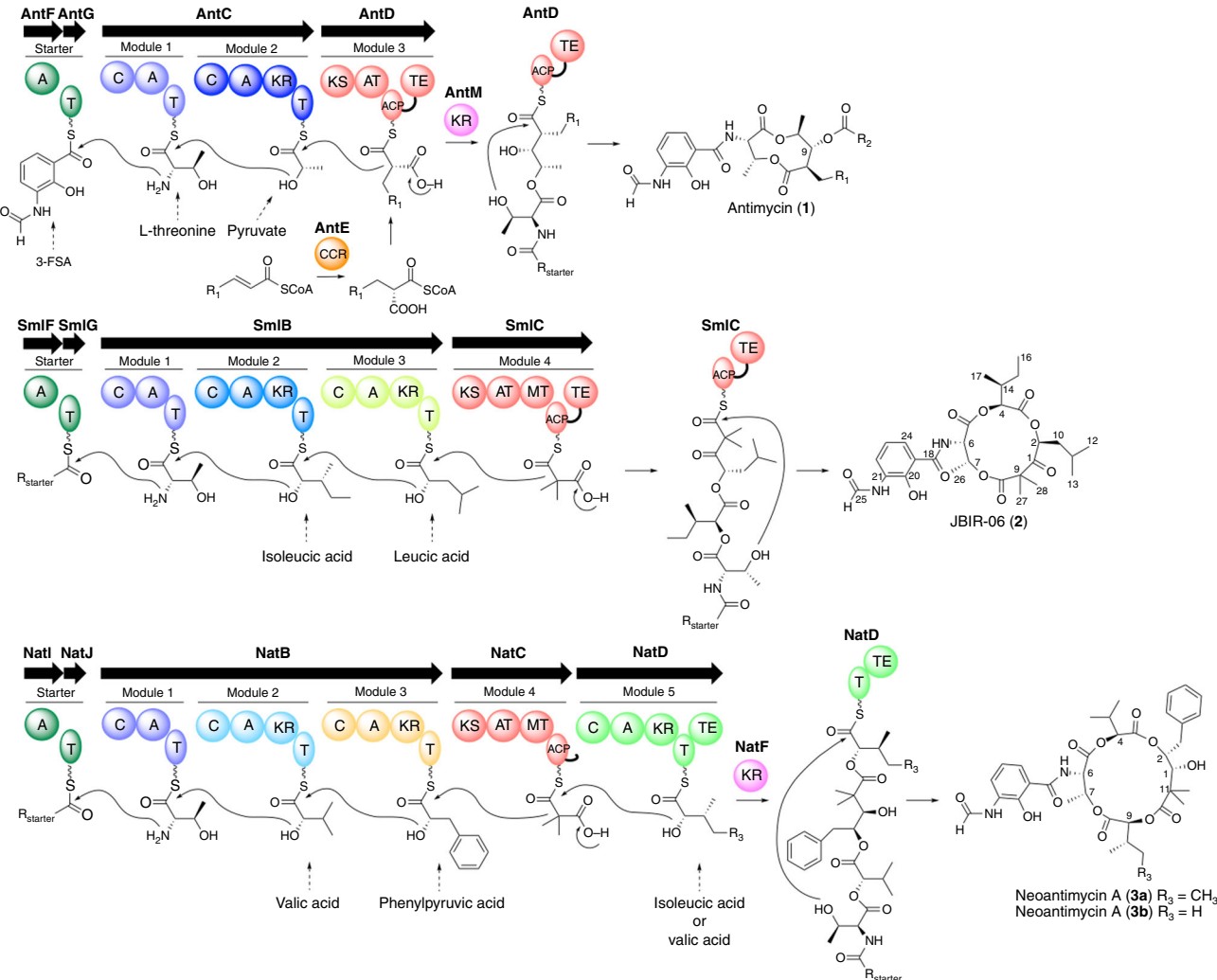

**Fig. 1** Organization of modular enzymes involved in the biosynthesis of antimycin-type depsipeptides. C condensation, A adenylation, T thiolation, KR ketoreductase, KS ketosynthase, AT acyltransferase, ACP acyl carrier protein, MT methyltransferase, TE thioesterase, CCR crotonyl-CoA reductase/carboxylase, $R_1$ and $R_2$ various alkyl groups, $R_{starter}$ the acyl moiety of 3-FSA. The module and domain organizations of the starter, the module 1, and the module 2 are identical among the all three systems. The starter and the module 1 uptake 3-FSA and L-threonine, respectively, common to all pathways, but the module 2 uptake pyruvate (AntC), isoleucic acid (SmlB), and valic acid (NatB), respectively. JBIR-06 and neoantimycin systems include an additional NRPS modules (module 3) after the module 2 in the same ORF. The module 3 in SmlB uptakes phenylpyruvic acid, and the module 3 in NatB uptakes leucic acid. The PKS modules of JBIR-06 and neoantimycin systems (module 4) contain MT domain between AT and ACP, and accept malonyl-CoA as an extender unit, to yield dimethyl group at the α-position of polyketide moiety, differently from antimycin system. Furthermore, neoantimycin system includes an extra NRPS module (module 5 as NatD) which uptakes isoleucic acid or valic acid

recombinants revealed that *S. lividans*/pKU518J06 and *S. lividans*/pKU518nant yielded **2** and **3a–3b**, respectively (Supplementary Fig. 3). Through NMR and MS analyses, **2**, **3a**, and **3b** were identified as JBIR-06, neoantimycin A, and neoantimycin F (Supplementary Figs. 4–7, Supplementary Table 3), by comparisons with data in the literature[28,36,37]. These results linked **2** to the product of *sml* cluster, and **3a–3b** to those of *nat* cluster, and provided the platform for the engineered biosynthesis.

**Bioinformatic analyses of the NRPS-PKS module structures.** Sequence analyses revealed that *sml* and *nat* NRPS/PKS gene clusters encode four and five modules, respectively, to build up the molecular scaffold of the tri-lactone JBIR-06 and the tetra-lactone neoantimycin (Fig. 1). We conducted the DNA sequence alignment for A, C, T, and KR domains from *ant*, *sml*, and *nat* system, respectively, and calculated their identities (Supplementary Fig. 8). Due to the high amino acid sequence

identities (i.e., the PKS modules AntD and SmlC share 51% amino acid sequence identity, while AntD and NatC share 53%), we could precisely identify functional domain boundaries, and thus intervening linkers or adjacent docking domains (Fig. 2a, Supplementary Fig. 9). It is critical to retain the intimate domain/module interactions that are required for precise control of the chain transfer and elongation reactions for successful engineering of the NRPS/PKS assembly lines. As a result, as shown in Fig. 2a, we annotated region 1 (1301–1371 in SmlC and 1320–1390 in NatC) as the ACP domain of PKS module, region 2 (1393–1647 in SmlC), which is only conserved in AntD and SmlC, as the TE domain, and the region surrounded by ACP and TE (1372–1392 in SmlC) as an inter-domain linker (Fig. 2a). Homology modeling (SWISS-MODEL server, https://swissmodel.expasy.org) supports the hypothesis on its structure including region 1 forms four α-helices and two loops, which are involved in the interactions with the KS-AT domains[38,39] (Fig. 2b, Supplementary Fig. 10). The C-terminal

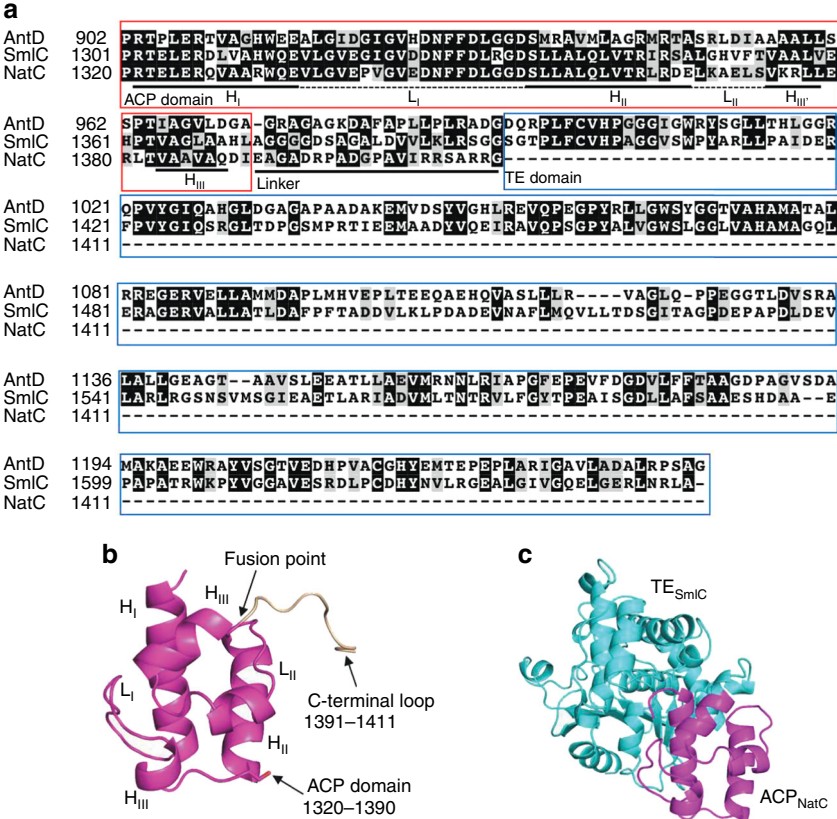

**Fig. 2** Bioinformatic analyses of the antimycin PKS modules. **a** Alignment of the PKS modules, AntD, SmlC, and NatC. We annotated $SmlC_{1301-1371}$ as an ACP domain (region 1), $SmlC_{1372-1392}$ as an interdomain linker, $SmlC_{1393-1647}$ as a TE domain (region 2), and $NatC_{1391-1411}$ as a docking domain. **b** A homology model of $ACP_{NatC}$ (1320–1411), based on the reported crystal structure of the surfactin synthase ACP subunit (PBD: 2VSQ) as the template. **c** A homology model of the $ACP_{NatC}$-$TE_{SmlC}$ domains, the NRPS module from *Acinetobacter baumannii* (NCBI PBD ID 4ZXH_A, https://www.ncbi.nlm.nih.gov/protein/4ZXH_A) was used as the template

region of NatC (1391–1411) was predicted to be a docking domain between PKS and NRPS modules, as the one previously reported for the epothilone NRPS-PKS system[40]. Therefore, with the bioinformatic information in our hands, we anticipated that the deletion of the NatD NRPS module from the tetra-lactone neoantimycin assembly line would lead to ring contraction (Fig. 3a), whereas the addition of the NatD module to the tri-lactone JBIR-06 system would expand the ring structure (Fig. 4a). Furthermore, in order to maintain the intimate domain/module interactions, we employed the TE domain of the SmlC PKS module ($TE_{SmlC}$), as well as the predicted interdomain linker between the ACP and TE domains of SmlC, for the ring contraction, while the docking domain between the NatC PKS module and the NatD NRPS module was kept for the ring expansion experiment. We therefore took advantage of this information to redesign the chimeric NRPS-PKS assembly lines for the ring contraction and expansion, as described below.

Notably, our sequence analysis of the C-terminal region of NatC revealed that $NatC_{1391-1411}$ contains several positively charged residues while other C-terminal docking domains that interact with NRPS subunits contain a run of negatively charged residues (Supplementary Fig. 11A). Further, the three α-helices and two β-turns, annotated as a docking domain in the EpoB NRPS structure[9], are uniquely replaced with a shorter 9 amino acid sequences in NatD (Supplementary Fig. 11B). This suggests that the docking between NatC and NatD is fundamentally different from other interfaces. Despite these dissimilarities, our

engineering strategy with the docking domain of NatC worked (see the section of ring expansion).

**Ring contraction of neoantimycin A**. We first conducted the ring contraction of the tetra-lactone neoantimycin A (**3a**), by deleting the NatD NRPS module from *nat* cluster and connecting the linker-$TE_{SmlC}$ domain right after the C-terminus of NatC, to form a NatC-SmlC chimeric PKS (Figs. 2c, 3a). TE domains usually exhibit tolerant substrate specificities, as exemplified by the early studies where the TE domain from erythromycin system was shown to accept polyketide chains of diverse lengths[41], and recognize the functional group at the acyl terminus and adjacent to the attacking nucleophile[42]. Therefore, we expected that $TE_{SmlC}$ domain, which recognizes the acyl group derived from polyketide and the hydroxyl group of threonine as a nucleophile, would accept the intermediate from the new construct and catalyze macrocyclization. The interdomain linker of SmlC was also employed, to connect the $ACP_{NatC}$ domain to the $TE_{SmlC}$ domain (Fig. 2a). The *S. lividans* strain harboring pKU518nantΔ*natD*::*smlCTE*, accordingly engineered with the Red/ET system[43] (Supplementary Note 2, Supplementary Tables 1–2), was incubated in A3M medium, and the extract of the transformant was analyzed by HPLC. As expected, the production of **3a**–**3b** was completely abolished, and **4** newly appeared (Fig. 3b). LC-MS analyses showed that the *m/z* of **4** was 581, which is 116 less than **3a**, and its UV max was around 315 nm, characteristic of the antimycin starter unit (Supplementary Figs. 4, 12, Supplementary

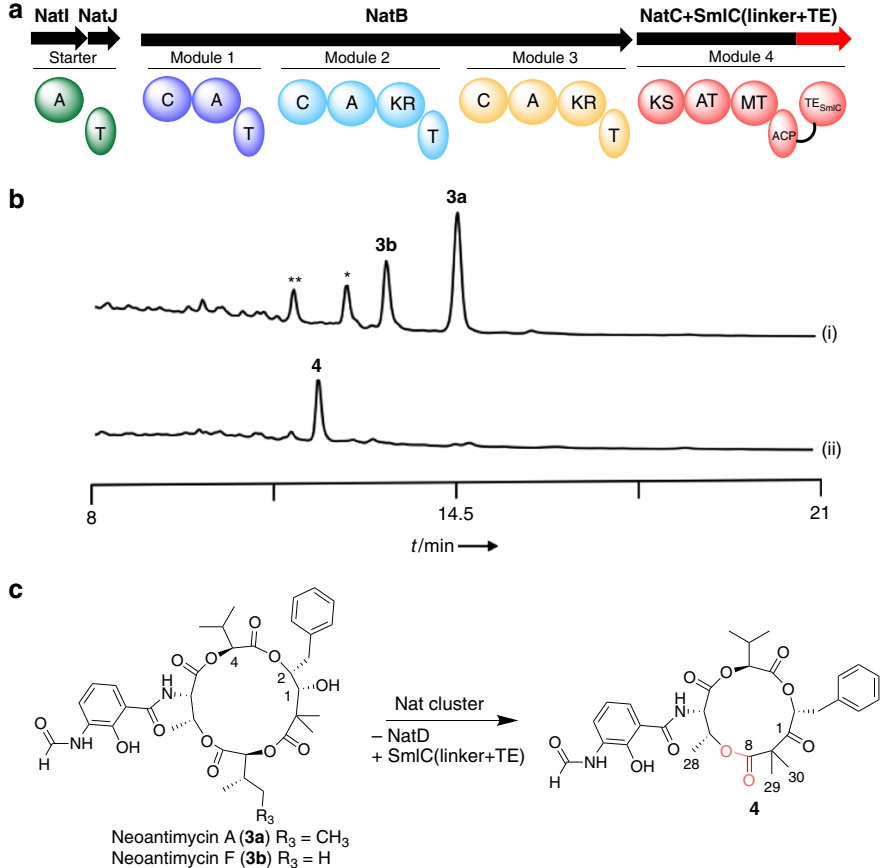

**Fig. 3** Ring contraction of neoantimycin A. **a** The reconstructed module structure for ring contraction. **b** HPLC analyses of the metabolites from transformants harboring (i) pKU518nant and (ii) pKU518nantΔ*natD::smlCTE*. The chromatogram represents the UV absorbance at 320 nm. * and ** indicate unidentified analogs with *m/z* values that were 14 and 28 less than **3b**, respectively. **c** Engineering scheme and structure of **4**

Table 3), suggesting that **4** was generated from the ring contraction of **3a**. The [1]H and [13]C NMR data defined the structure of the side chain, and the HMBC correlations from H-28, H-29, and H-30 to C-8 (Supplementary Figs. 13–18, Supplementary Table 4) clearly showed that the hydroxyl group of the L-threonine moiety was esterified with the carboxyl group of the polyketide moiety, indicating that **4** is a trilactone generated by the loss of isoleucic acid from **3a** (Fig. 3c). **4** bears a keto group at C-1, indicating that the ketoreduction unexpectedly did not work in the recombinant strain. It might be caused by the polar effect on the expression of *natF*, a ketoreductase gene located downstream of *natD*, though we cannot eliminate the possibility that NatF no longer recognizes the substrate. The yield of **4** (3.9 ± 0.7 mg/L) was 3-fold reduced relative to that of **3a** (12 ± 2.0 mg/L) (Table 1), but it did not drop significantly, verifying the accuracy of our domain structure prediction.

**Ring expansion of JBIR-06.** We next enlarged the lactone size of the tri-lactone JBIR-06 (**2**) by creating an *sml/nat* chimera. To this end, we appended the NatD NRPS module to *sml* cluster and replaced the linker and TE domain in the SmlC PKS module with the docking domain of NatC (Fig. 4a, Supplementary Fig. 6A), to maintain the module/module interactions to yield the tetra-lactone product. The accordingly engineered BAC, pKU518J06ΔsmlCTE (Supplementary Note 3, Supplementary Tables 1–2), was introduced into *S. lividans* with the *natD* expression vector, based on the φC31 phage integration system (pZH2-NatD) (Supplementary Note 4, Supplementary Fig. 19). As a result, the original product **2** disappeared completely, and **5**

and **6** newly appeared in the transformant (Fig. 4b). The planar structure of **5** was determined as the tetra-lactone, which contains one threonine, two isoleucic acids, and one leucic acid (Fig. 4c), through a comparison of the 1D NMR data with the literature (Supplementary Figs. 20–23, Supplementary Table 5)[27]. The HMBC correlations between H-9 and C-8/-10, and H-7 and C-8 and MS data indicated that the additional isoleucic acid was introduced into the C-terminus of the depsipeptide chain of **2**, and lactonized with the hydroxyl group of threonine to yield **5** (Supplementary Figs. 4, 18, and 24, Supplementary Table 3). The stereochemistry of each building block was identified as L-threonine, L-isoleucic acid, and L-leucic acid with the chiral GC-MS analyses of the acid-hydrolyzed fragments of **5** and **6**, and the linear depsipeptide consisted of 3-FSA starter unit, threonine, isoleucic acid, and leucic acid (Supplementary Figs. 25–30, Supplementary Table 6). The yield of **5** (5.9 ± 1.6 mg/L) was almost the same as that of **2** (5.9 ± 0.7 mg/L, from *S. lividans*/ pKU518J06) (Table 1), indicating that NatD accepted the intermediate from SmlC efficiently and esterified one more L-isoleucic acid with the intermediate, to produce the tetra-lactone **5**. The unexpected accumulation of the acyclic **6** (9.7 ± 0.8 mg/L) might have been caused by the extended residence time of the intermediate on SmlB relative to the native system, leading to its spontaneous hydrolysis.

**Alkyl chain diversification of JBIR-06.** As described above, both SmlC and NatC PKS modules include an additional methyltransferase (MT) domain, which is not present in the antimycin system (Fig. 1). In addition, the *sml* and *nat* clusters both lack the

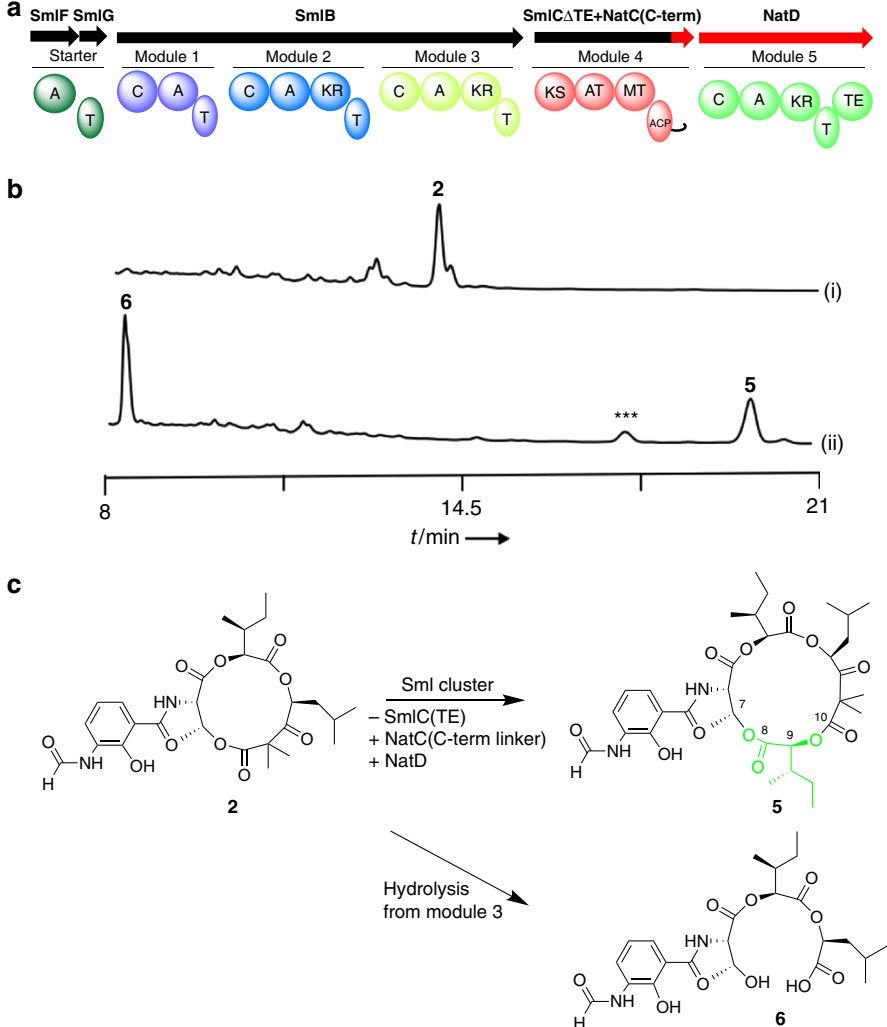

**Fig. 4** Ring expansion of JBIR-06. **a** The reconstructed module structure for ring expansion. **b** HPLC analyses of the metabolites from transformants harboring (i) pKU518J06 and (ii) pKU518J06ΔsmlCTE/pZH2-NatD. The chromatogram represents the UV absorbance at 320 nm. *** indicates a mixture of analogs with $m/z$ values that are 14 less than **5**. **c** Engineering scheme and structure of **5** and **6**

CCR enzyme[20–23], which supplies various alkylmalonyl-CoA extender substrates to generate the alkyl group variations. Unlike the antimycin system, the JBIR-06 and neoantimycin systems lost the alkyl chain variations and only form the dimethyl group at C-9 in JBIR-06 and at C-11 in the neoantimycins, respectively (Fig. 1). Therefore, in order to increase the structural variations of the medicinally important JBIR-06, we planned to utilize the previously reported engineered CCR, the structure-based mutant of AntE that can supply a wide range of longer and bulkier alkylmalonyl-CoAs[22], which is not straightforward by organic synthesis. Importantly, the long-chain alkyl group in the antimycin-type depsipeptide scaffold is reportedly essential for the Bcl2/Bcl-xL inhibitory activity, which leads to cancer cell death[32,44]. At the same time, we also engineered SmlC PKS module by introducing mutations into the AT domain in order to broaden its substrate specificity toward that of the antimycin system (Fig. 5a), so that the engineered PKS can process the unnatural extender units. Thus, we introduced SmlCAT_ant, in which the substrate recognition motif "LRIAPH" was substituted with MPAAAH from AntD (Supplementary Fig. 9, Supplementary Note 5, Supplementary Tables 1–2), into *S. lividans* harboring pKU518J06ΔSmlC, with *smlC* deleted (Fig. 5a, Supplementary Note 6). We focused on this motif since it has

been reported that the mutation of this site VDYASH in the AT domain of the erythromycin PKS altered the substrate preference[45–48].

The introduction of AntEV350G and SmlCAT_ant gave rise to the production of **7a–7f**, with $m/z$ values that are 28 (**7a–b**), 42 (**7c–e**), and 56 (**7f**) higher than **2** (Supplementary Fig. 4), suggesting that they possess longer alkyl chains (Fig. 5b). Using NMR and HR-MS data, we characterized **7a**, **7c**, and **7f** as JBIR06-type tri-lactone depsipeptides with butyl (**7a**), pentyl (**7c**), and hexyl (**7f**) chains at the C-9 position (Fig. 5c, Supplementary Figs. 4, 31–45, Supplementary Tables 3 and 7–9). Compounds **7b**, **7d**, and **7e** were also identified as their minor derivatives (Supplementary Figs. 46–51, Supplementary Table 10). Small amounts of **7a–7f** were observed in the strain harboring pKU518J06 with AntEV350G, but these yields were 2–3 times lower than that of the AT_SmlC-mutated construct (Table 1). These data suggested that the mutation into AT_SmlC broadens the substrate specificity to accept a variety of extender substrates supplied by AntEV350G. Notably, the C-9 di-substituted pattern of **2** changed into a mono-substitution, suggesting that the MT domain of SmlC does not work when the longer alkyl chain has been incorporated. However, it should be noted that the yield of the native product **2** is still dominant (5.0 ± 1.7 mg/L) (Table 1),

**Table 1 The yields of isolated products**

| Product | Yields[a] (mg/L) |
|---|---|
| 3a | 12 ± 2.0 |
| 3b | 6.5 ± 1.4 |
| 4 | 3.9 ± 0.7 |
| 2[b] | 5.9 ± 0.7 |
| 5 | 5.9 ± 1.6 |
| 6 | 9.7 ± 0.8 |
| 2[c] | 5.0 ± 1.7 |
| 7a[c] | 1.2 ± 0.2 |
| 7b[c] | 0.35 ± 0.06 |
| 7c[c] | 0.78 ± 0.17 |
| 7d[c] | 0.33 ± 0.07 |
| 7e[c] | 0.39 ± 0.13 |
| 7f[c] | 1.4 ± 0.3 |
| 2[d] | 4.1 ± 0.8 |
| 7a[d] | 0.53 ± 0.06 |
| 7b[d] | 0.15 ± 0.02 |
| 7c[d] | 0.18 ± 0.02 |
| 7d[d] | 0.11 ± 0.02 |
| 7e[d] | 0.05 ± 0.01 |
| 7f[d] | 0.42 ± 0.02 |

[a]The quantification was done based on the standard curve using **6** ($n = 3$, mean ± SEM)
[b]From *S. lividans* harboring pKU518J06
[c]From *S. lividans* harboring pKU518J06ΔSmlC/pZH2-SmlCAT$_{ant}$-AntEV350G
[d]From *S. lividans* harboring pKU518J06ΔSmlC/pZH2-AntEV350G

indicating further optimization is required. Nonetheless, it was remarkable that the total yield of the newly obtained JBIR-06 derivatives with different alkyl chain structures, **7a**, **7c**, **7e**, and **7f** (3.8 mg/L), is almost comparable with that of **2**.

## Discussion

In this study, we accomplished three manipulations of the antimycin-type NRPS-PKS assembly lines and obtained nine depsipeptides (**4**, **5**, **6**, **7a–f**) with different lactone ring sizes in substantial yields. It is quite remarkable that the yields of the compounds which we obtained in the engineered NRPS-PKS system are 5–10 times higher than those in the reported module assembly line engineering[11,49–51]. For the ring reduction approach, the linker between the ACP and TE domains smoothly connected the TE$_{SmlC}$ domain to the ACP$_{NatC}$, resulting in the production of the tri-lactone **4** without significant drop of yield. Since TE$_{SmlC}$ macrocyclizes a tri-lactone compound which shares the same structures around the carboxyl and attacking hydroxyl groups as the original substrate, we anticipated that TE$_{SmlC}$ would work efficiently in the engineered system. In this approach, we reversely traced the evolutionary flow from the Sml to Nat modules, which facilitated the efficient reconstruction without large changes in the intermediate structures.

In the ring expansion approach, the intersubunit docking domain at the C-terminus of the NatC module was added onto the SmlC domain, instead of the TE domain. Remarkably, the yield of the ring-expanded compound **5** was same degree as that of original products **2**. This data demonstrated that our strategy was effective to newly generate the module interactions. The interactions between NatC and NatD are likely to be different from the known systems, because the N-terminus of NatD lacks a predicted secondary structure consistent with the crystal structure of the docking domain from the EpoB NRPS[9]. The interactions between PKS and NRPS modules are still elusive, and need to be clarified by X-ray crystallization or cryo-EM analyses for future engineering studies.

In our alkyl chain diversification approach, we introduced the broad substrate specificity of AntD into SmlC by mutating the AT

substrate definition sequence. The substrate definition sequence "M$_{701}$PAAAH$_{706}$" in AntD is exceptionally simple, and it likely causes the expansion of substrate binding cavity. In fact, the two alanines corresponding to A$_{703}$ and A$_{705}$ were recently shown to be the residues that influence substrate specificity of SpnD-AT[52]. In the case of the systems with relaxed substrate specificity for acyl-CoAs, the CCR enzyme family[20–23] could be used to diversify product structures in future. The butyl-, 3-methylbutyl, and hexylmalonyl-CoAs provided by AntEV350G for **7a**, **7c**, and **7f** production were likely synthesized from 2,3-hexenoyl, 5-methyl-2,3-hexenoyl, and 2,3-octenoyl-CoAs. This side chain pattern was also observed in our previous engineering studies for **1**[23], implying the wide distribution of 2,3-alkenoyl-CoA in *Streptomyces*. As the original product (**2**) was still dominant in our system, the expansion of extender unit specificity of KS domain should be effective to increase the yields. With expecting the higher yield, we also tested several chimeric PKS module constructs such as KS$_{SmlC}$-AT$_{AntD}$-MT ACP TE$_{SmlC}$ with domain swapping strategy (Supplementary Fig. 52); however, none of them afforded any detectable products (data not shown). Further adoption of the recently identified "optimal" fusion junctions for AT-domain swaping[53] may improve the product yields.

Consequently, this study paves the way for the rational engineering of NRPS-PKS assembly machineries, by following the evolution of enzymes in nature. Through sequence comparisons, we can learn how nature cut and paste module structures. In addition, by employing the enzymes to give the various functional groups, we can further increase the diversity of products, as we have done previously with CCR enzymes[23]. The *Escherichia-Streptomyces* shuttle BAC genetic platform that we used is also an important factor to readily build artificial biosynthetic modules. The productivity of the engineered modules would be improved by optimizing the interactions betwen domains and modules based on the structural analysis of the whole module complex, even though it still remains a technically challenging endeavor. This study confirmed that the strategies applied to pure PKS and NRPS systems can be productively used for hybrid PKS/NRPS system. Furthermore, this study serves as an exemplar for bioengineering productive biosynthetic assembly lines that produce unnatural polyketide-non-ribosomal peptides.

## Methods

**General experimental procedures**. Solvents and chemicals were purchased from Wako Chemicals Ltd. (Tokyo, Japan) or Kanto Chemical Co., Inc. (Tokyo, Japan), unless noted otherwise. Oligonucleotide primers were purchased from Eurofins Genetics (Tokyo, Japan) and Sigma-Aldrich Japan (Tokyo, Japan). PCR was performed using a TaKaRa PCR Thermal Cycler Dice® Gradient (TaKaRa), with Prime Star Max (Takara). Sequence analyses were performed by Eurofins Genetics (Tokyo). Analytical and preparative HPLC were performed on a Shimadzu Prominence system. Silica gel column chromatography was performed using Wakogel C-200. NMR spectra were obtained at 500 MHz ($^1$H) and 125 MHz ($^{13}$C) with a JEOL ECX-500 or ECZ-500 spectrometer, and chemical shifts were recorded with reference to solvent signals ($^1$H NMR: CDCl$_3$ 7.26 ppm; $^{13}$C NMR: CDCl$_3$ 77.0 ppm). All NMR spectra were measured by using CDCl$_3$ as solvent. Samples for LC-MS analysis were injected into an Shimadzu Prominence system HPLC-MicroTOF mass spectrometer (Bruker Daltonics), using electrospray ionization with a COSMOSIL 2.5C$_{18}$-MS-II column (2.0 i.d. × 75 mm; Nacalai Tesque, Inc.).

**Genome sequencing**. The complete genome sequence of *Streptomyces* sp. ML55 and *Streptomyces orinoci* NBRC13466 was determined by using a Miseq (Illumina, San Diego, CA, USA) and a PacBio RS II (Pacific Biosciences, Menlo Park, CA, USA). End sequencing was carried out using the BigDye terminator ver3.1 kit (Applied Biosystems). The obtained sequence data were assembled using HGAP2 (Pacific Biosciences).

**Analytical conditions**. *S. lividans* transformants were cultured at 30 °C for 4 days in 500 mL flasks containing 100 mL A3M medium (Glucose 0.5%, Glycerol 2%, Soluble Starch 2%, Pharmamedia 1.5%, Yeast extract 0.3%, HP20 1%, pH = 7.0). After filtering, the mycelia were lyophilized to dryness. Dried mycelia were extracted by MeOH:CHCl$_3$ = 2:1. The extraction solution was concentrated in

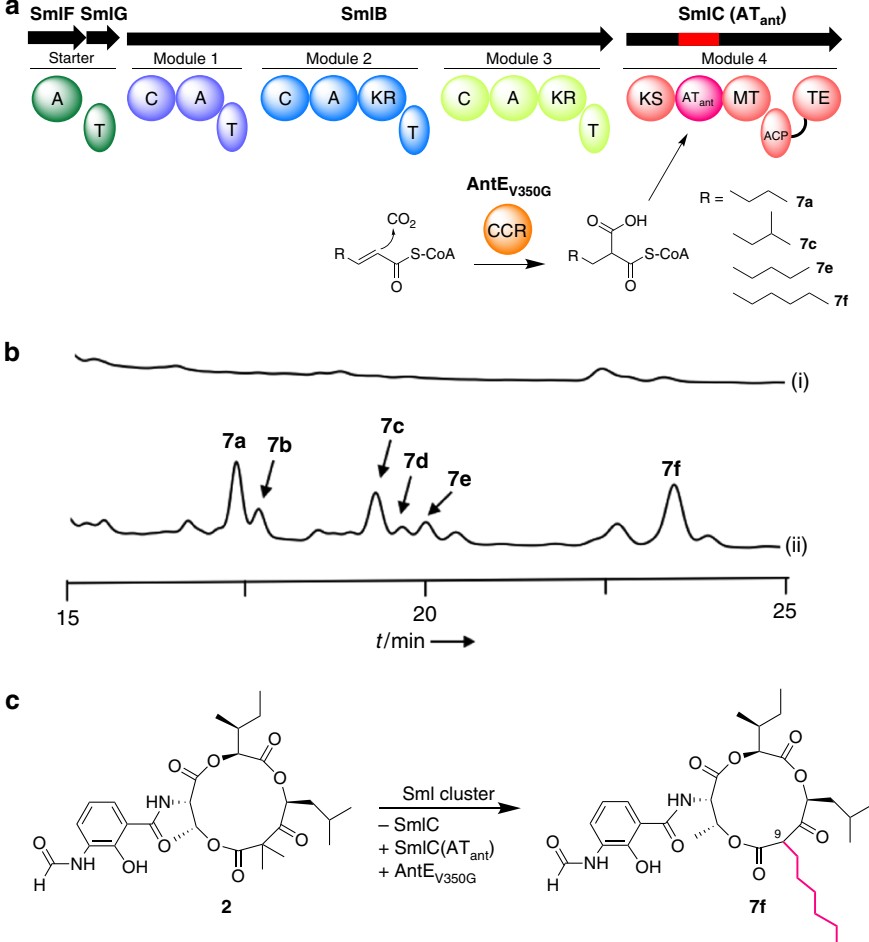

**Fig. 5** Alkyl chain diversification of JBIR-06. **a** The reconstituted module structure for alkyl chain diversification. **b** HPLC analysis of the metabolites from transformants harboring (i) pKU518J06 and (ii) pKU518J06ΔSmlC/pZH2-SmlCAT$_{ant}$-AntEV350G. The chromatogram represents the UV absorbance at 320 nm. **c** Engineering scheme and structure of the representative product **7f**

vacuo to remove the solvent. Products from *S. lividans* TK21 transformant were analyzed by HPLC equipped with a STAR Separar C18G 5 μm column (4.6 i.d. × 250 mm, Rikaken Co. Ltd., Nagoya, Japan), with a solvent system of 0.1% formic acid (solvent A) and acetonitrile containing 0.1% formic acid (solvent B), at a flow rate of 1.0 mL/min and a column temperature of 40 °C. Separation was performed with solvent B/solvent A (50:50), a linear gradient from 50:50 to 80:20 within the following 5 min, an isocratic elution with 80:20 within the following 25 min.

**Isolation of the metabolites from *S. lividans* TK21**. *S. lividans*/pKU518nantΔ*natD*::*smlCTE* was cultured at 30 °C for 4 days in 30 × 500 mL flasks containing 100 mL A3M medium (Glucose 0.5%, Glycerol 2%, Soluble Starch 2%, Pharmamedia 1.5%, Yeast extract 0.3%, HP20 1%, pH = 7.0). After filtering, the mycelia were lyophilized to dryness. Dried mycelia were extracted by MeOH:CHCl₃ = 2:1. The extraction solution was concentrated in vacuo to remove the solvent. The extract was subjected to silica-gel column chromatography and eluted using chloroform 100%. *S. lividans*/pKU518J06Δ*smlCTE*/pZH2-NatD was cultured at 30 °C for 4 days in 5.0 L A3M medium, and *S. lividans*/pKU518J06Δ*smlC*/pZH2-SmlCAT$_{ant}$ was cultured at 30 °C for 4 days in 12 L A3M medium. The compounds were extracted and purified by silica-gel as described above.

Fractions containing **4** was further purified by reverse-phase preparative HPLC equipped with an 5C18-MS-II column (Nacalai Tesque, Kyoto, Japan, 5 Δm, 10 mm i.d. × 250 mm). Separation was performed with solvent B (CH₃CN)/solvent A (0.05% formic acid) (70:30), a linear gradient from 70:30 to 85:15 within the following 10 min, a linear gradient from 85:15 to 100:0 within the following 0.5 min, 100:0 for 4.0 additional min, to yield **4** (1.1 mg). Fractions containing **5** and **6** were further purified by preparative HPLC equipped with a YMC-Triant C18 column (YMC, Kyoto, Japan, 5 μm, 10 mm i.d. × 250 mm) using acetonitrile-0.05% formic acid (80:20) as the eluting solvent (flow rate 1.0 mL/min), to yield **5** (2.1 mg) and **6** (2.2 mg). Fractions containing **7a–f** were further purified by preparative HPLC equipped with an YMC-Triant C18 column (YMC, Kyoto, Japan, 10 μm, 10 mm i.d. × 250 mm) using acetonitrile-0.1% formic acid (80:20) as the eluting solvent (flow rate 3.0 mL/min), and further purified by preparative

HPLC equipped with an X-select HSS T3 column (Waters, MA, USA, 10 μm, 10 mm i.d. × 250 mm) using acetonitrile-0.1% formic acid (70:30) as the eluting solvent (flow rate 3.0 mL/min) to yield **7a** (2.4 mg), **7b** (0.6 mg), acetonitrile-0.1% formic acid (72.5:27.5) to yield **7c** (1.0 mg), **7d** (0.2 mg), **7e** (0.3 mg), and acetonitrile-0.1% formic acid (75:25) to yield **7f** (2.9 mg).

**Chemical synthesis of isoleucic/leucic acids**. L-isoleucine (100 mg) was dissolved in 1.25 M H₂SO₄ (5 mL) and stirred on ice, and ice-cold aqueous solution of NaNO₂ (0.2 g/mL, 4 mL) was slowly added and stirred for 2 h on ice. Then, the reaction mixture was moved to room temperature, and further stirred for 15 h at room temperature. Finally, the reaction mixture was directly extracted with diethyl ether (5 mL × 2), and organic layer was dried with Na₂SO₄. The organic solvent was removed under reduced pressure and L-isoleucic acid was obtained as colorless oil (56 mg). Other isoleucic/leucic acids were also prepared from corresponding amino acids by diazotization except for L-leucic acid (Wako Pure Chemical Industries, Ltd.).

**Methyl esterification of isoleucic/leucic acids**. A methanol solution of L-isoleucic acid (50 mg/10 mL) in eggplant flask was stirred at room temperature. Then, thionyl chloride (1.2 mL) was slowly added to the solution. The mixture was further stirred for 2 h at 100 °C (in reflux), and cooled to room temperature. The solvent was removed under reduced pressure and the residue was suspended in diethyl ether (10 mL). The suspension was washed with saturated sodium hydrogen carbonate (10 mL × 2), and the organic layer was dried with Na₂SO₄. Finally, the ether was removed by evaporation and the L-leucic acid methyl ester was obtained as colorless oil (42 mg). For other isoleucic/leucic acid methyl esters were also prepared in the same procedure.

**Preparation of N-TFA-threonine methyl esters**. L-threonine (1 mg) was dissolved in hydrogen chloride-methanol reagent (5–10%) (0.5 mL, purchased from Tokyo Chemical Industry Co., Ltd.), and the mixture was incubated for 30 min at

100 °C (in reflux). The solvent was removed under reduced pressure. To the residue, a 1:1 mixture of trifluoroacetic anhydride (TFAA)/dichloromethane (0.5 mL) was added and the mixture was incubated for 30 min at 100 °C (in reflux). After the reaction, the solvents were removed by argon gas injection. Finally, the residue was dissolved in acetone (0.5 mL) and subjected to GC-MS analysis as a standard. The other stereoisomers of threonine were also derivatized in the same procedure and subjected to GC-MS analysis.

**Hydrolysis of 5 and 6**. Compound **5** (0.1 mg) was added to 6 M hydrochloric acid (0.5 mL) and stirred for 24 h at 110 °C (in reflux). After the reaction, the mixture was cooled on ice to room temperature, and the solvent was removed by lyophilization. The resulting residue was used for further chemical derivatization as noted below. Compound **6** was also hydrolyzed by the same procedure.

**Chemical derivatization of hydrolysates of 5 and 6**. The hydrolysate of **5** (from 0.1 mg) was in hydrogen chloride-methanol reagent (5–10%) (0.5 mL), and the mixture was incubated for 30 min at 100 °C (in reflux). The reaction mixture was divided in two equal parts, and the solvent was removed under reduced pressure. The half part of the residue was dissolved in acetone (0.5 mL) and subjected GC-MS to analyze isoleucine/leucine acids as methyl ester. The remaining half part of the residue was dissolved in the mixture of TFAA/dichloromethane (0.5 mL), and the mixture was incubated for 30 min at 100 °C (in reflux). The solvents were removed by argon and the residue was dissolved in acetone (0.5 mL), which was subject to the GC-MS to analyze threonine moiety as the N-TFA methyl ester. The hydrolysate of **6** (from 0.1 mg) was also derivatized and subjected to the GC-MS analysis in the same procedure.

**GC-MS analysis condition**. CP-Chirasil Dex-CB column (Alltech, 0.25 mm × 25 m; He as the carrier gas; program rate: 50–150 °C at 5 °C/min, 150–200 °C at 50 °C/min).

**GC-MS for threonine derivatives**. CP-Chirasil Dex-CB column (Alltech, 0.25 mm × 25 m; He as the carrier gas; program rate: 50–200 °C at 6 °C/min).

**Data availability**. The DNA sequences of JBIR-06 and neoantimycin biosynthetic gene clusters were registered as LC375135 and LC375136 in DDBJ, respectively.

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

## Acknowledgements

This work was supported by a Grant-in-Aid for Scientific Research from the Ministry of Education, Culture, Sports, Science and Technology, Japan (JSPS KAKENHI Grant Number JP15H01836, JP16H06443, JP16K13084, and JP17H04763), JST/NSFC Strategic International Collaborative Research Program Japan-China, Kobayashi International Scholarship Foundation, and National Natural Science Foundation of China Grants (21520102004). We also thank Prof. Hiroyasu Onaka for providing genetic tools.

## Author contributions

T.A., L.Z., W.L., K.S. and I.A. designed the experiments. T.A., T.F., L.Z., S.H., Z.H., J.H., I.K. and H.I. performed the experiments. T.A., T.F., L.Z. and I.A. analyzed the data. T.A., K.S. and I.A. wrote the paper.

## Additional information

**Competing interests:** The authors declare no competing interests.

