## [Peer Review File · Nature Communications]

Reviewers' comments:

Reviewer #1 (Remarks to the Author):

In this manuscript, Awakawa et al. describe (re)identify the biosynthetic gene clusters for the antimycin-type depsipeptides JBIR-06 and neoantimycin and establish a heterologous production platform for these compounds using *Streptomyces lividans*. The authors used amino acid alignments to identify amino acid signatures delineating key points in the assembly line (i.e. linkers to the TE domain and for interacting with the upstream module). They then used this information to design a genetic strategy for addition or subtraction of a module from the JBIR-06 and neoantimycin biosynthetic systems, respectively, and demonstrate that the new system produces the expected products and verify these by high HR MS and NMR. Finally, the authors introduced codons specifying AAAH into the JBIR-06 PKS at canonical substrate utilization signature motif III (which presumably enable the broad substrate utilization observed for the antimycin PKS) and observed the corresponding changes in alkylation at C9.

Progress in bioengineering natural products biosynthesis, particularly with regard to making major changes to the core chemical scaffold, has lagged considerably. Most bioengineering efforts have focused on making small changes in post-assembly line tailoring steps in part because there is a lack of structural detail about the communication occurring between modules and domains. The authors' strategy to use 'evolution' as the foundation for the engineering strategy is indeed a clever one. I read this manuscript with some interest and overall found it to be an exciting piece of work that I think makes a significant contribution to the field of natural products bioengineering. However, I do have a few comments that should be addressed, mostly regarding the discussion section, which I think reads more like an article summary section and requires flushing out.

Comments:

1-- In the first results section, where are the HPLC traces or EICs demonstrating that TK21 harbouring pKU518J06 and pKU518nant can produce JBIR-06 and neoantimycin, respectively? These are visible in the Fig 3a and Fig 4a, but there should be a standalone figure for this even if it is only in the supplemental information.

2-- Please elaborate what is meant by, "we confirmed that these module structures were the same as those in the previous report, except for NatD".

3-- The amino acid alignment and assembly line for the ring contraction experiment are mistaken called out in this sentence: "HPLC analyses of the culture extracts of the recombinants revealed that *S. lividans*/pKU518J06 and *S. lividans*/pKU518nant yielded 2 and 3a-3b, respectively (Figs. 2a and 3a)".

4-- The supplemental figures containing the NMR and MS analyses establishing the identity of compounds 2, 3a and 3b need to be called out.

5-- The paragraph starting 'Notably, in JBIR-06 and neoantimycin biosynthesis...' does not really work well in this section. In my view, it would work better at the start of the AT domain engineering section later on.

6-- The construction of chimeric biosynthetic assembly lines in the supplemental information is a bit difficult to follow. It might be helpful to remove the primers from the text and put all of these in your primer table. It would also be useful to have a table of plasmids constructed and their use in the study.

7-- Concerning the sentence, "At the same time, we also engineered the SmIC PKS module by replacing the AT domain...". From what I can tell, the entire domain was not swapped, rather only

a few codons were changed. Please amend your language.

8-- The discussion section reads more like a summary than an actual discussion section. I appreciate that there is a word limit, but the authors need to more comprehensively put their data into context within the natural products bioengineering field. For instance, aside from the recent paper from Wlodek et al (which the authors do cite), there has been very little if any successful experiments where the 'size' of the chemical scaffold has been rationally engineered. This needs to shine through more. A broader discussion of linker domains and what is known would also seem to be warranted as well. For example, "The sequence of the intermodule inker of NatC was not similar to the one reported in the EpoA study³⁸" is not really meaningful to the reader. In addition, during the time this paper was in submission it would seem that structural data for the broad specificity of the antimycin PKS was published in *Angewandte Chemie*. This should be included in the discussion section and the text modified accordingly.

<https://onlinelibrary.wiley.com/doi/10.1002/anie.201802805>

Reviewer #2 (Remarks to the Author):

This paper reports the comparative analysis of three PKS-NRPS assembly lines encoding structurally-related metabolites, leading to the successful reconfiguration of two of them by genetic engineering. The initial achievement was to establish the heterologous expression of the target systems encoding the biosynthesis of JBIR-06 and neoantimycin, in the process confirming identities of the clusters suggested by others. The authors then modified the assembly lines as follows:

- i. neoantimycin: generation of a smaller derivative by removal of one chain extension cycle via relocation of a chain-terminating thioesterase to the penultimate module;
- ii. JBIR-06: addition of a chain extension cycle to generate a ring-expanded derivative, by grafting on a module using compatible docking domains;
- iii. JBIR-06: site-directed mutagenesis of the AT domain to accept novel extender units generated by co-expression in the strain of a previously-engineered broad specificity crotonyl-CoA carboxylase/reductase (ccr) enzyme AntEV350G, resulting in several new derivatives.

Although these experiments were all successful (the spectral data required to validate the structures are included) and the novel compounds obtained in quite respectable yields relative to the parent metabolites, the major criticisms against the work concern both the novelty of the engineering rationale, as well as the specific approaches taken. The argued basis for the experiments is the comparative analysis of three similar (and thus, presumably evolutionary-related) pathways, in order to define favorable sites for genetic engineering. However, this type of approach has in fact underpinned the bulk of genetic manipulations carried out on the so-called 'cis-AT' PKS systems, and is now routinely coupled with structural data obtained on individual domains to clearly define domain/linker/docking domain boundaries (this idea was recently codified in Eng, et al. (2018) *Nucleic Acids Res.* 46, D509, in which a computational platform was developed to aid in designing chimeric PKSs based on comparison of the sequences of PKS modules). Transplantation of a TE domain in order to effect early chain release is also well-documented (e.g. Cortes, et al. (1995) *Science* 268, 1487; Kao, et al. (1995) *JACS* 117, 9105; Martin, et al. (2008) *Org. Biomol. Chem.* 1, 4144). The experiment in which the AntE (ccr) is combined with mutagenesis of the SmIC AT domain is elegant and effective, but site-directed mutagenesis of ATs has already been demonstrated repeatedly to allow incorporation of novel extender units into intact systems (e.g. Bravo-Rodriguez, et al. (2015) *Chem. Biol.* 22, 1425; Sundermann, et al. (2013) *ACS Chem. Biol.* 8, 443; and several other papers from the same group). Also, even the native SmIC AT domain accepts the same extender units, giving the target compounds at yields only 2-3 fold lower than the mutated AT.

Together, these observations raise the question of whether successfully applying the same set of strategies to a hybrid PKS/NRPS is sufficiently innovative to merit publication in *Nat. Commun.*

That being said, the ring-expansion of JBIR-06 via docking domain engineering is notable, as analogous module insertion experiments have only been attempted previously by covalent fusion.

The manuscript has several other substantial weaknesses and a number of minor issues, as detailed below.

Further major comments:

1. Revisiting the 'evolutionary' justification for the modifications made, several other issues need to be considered. For cis-AT PKSs, the predominant model is one of repeated cycles of duplication of ancestral modules coupled with domain exchanges (Jenke-Kodama & Dittmann (2009) *Phytochemistry* 70, 1858), while for the distinct class of trans-AT PKSs, it is rather horizontal gene transfer that appears to dominate (Nguyen, et al. (2008) *Nat. Biotechnol.* 26, 225). In any case, the specific evolutionary history of hybrid cis-AT PKS/NRPS has yet to be rigorously investigated. In this context, the description of how PKSs have evolved (top of page 4) is too simplistic.

The authors then make specific reference (on page 4) to a revised model of evolution of cis-AT PKSs in which the modular unit is not as classically defined (i.e. running from KS->ACP), but from AT->KS. However, as they themselves pointed out in the earlier work (ref. 18), while this mode of evolution seems to have been followed for the set of closely-related polyene systems investigated in the study, it may not be generalizable to all cis-AT PKS systems. In any case, it is not particularly relevant to the experiments described here, as no module swaps were actually carried out according to the revised definition of a module.

The sentence starting 'This is a promising way to produce...' (top of page 4) is also problematic for a number of reasons, most notably because this paper doesn't address the 'reconstructing of domain structures', and because the predominant effect of recapitulating the native mode of evolution during engineering is not on 'substrate specificity' but on the structure of modules, as well as the protein-protein interactions which underlie their function.

Finally, the bioinformatics analysis described on page 5 runs counter to an analysis of the same PKSs published by others (ref. 32). In the present work, the authors propose that the third NRPS modules of the *sml* and *nat* clusters likely derived from the second module encoded by *antC* in the *ant* cluster, and similarly that the fifth module of the *nat* cluster probably evolved from the second or third module encoded by *smlB*. However, Magarvey et al. specifically argue based on their independent phylogenetic analysis, that neither of these events is likely. The discrepancy between these analyses needs to be explained, and the phylogenetic data supporting the evolutionary model proposed here need to be presented. What the availability of multiple apparently related systems instead allowed was a fine analysis of the module sequences in order to clearly identify functional domain boundaries, and thus intervening linkers or adjacent docking domains (although such identifications can also be performed, as pointed out earlier, with high success based on available crystal structures of the individual domains/docking domains – indeed, the authors themselves used homology modeling to refine their linker region assignment).

2. The authors need to be more rigorous in their use of PKS/NRPS-specific vocabulary. Sequences within PKS or NRPS polypeptides (i.e. intrasubunit) that connect functional domains and modules are referred to as 'linkers', while those that mediate communication between subsequent subunits (i.e. intersubunit) are referred to as 'docking domains' (see refs. 7, 9 and 10). Using these two terms in the same sentence (or indeed interchangeably (as in the discussion)) is confusing. In this context, the engineering of compound 5 by addition of a module, relied on manipulation of docking domains. Although this was carried out to some extent successfully (see below), it is surprising, given that the authors were not able to identify which type of docking domain is actually functioning at the NatC/NatD interface. They make an analogy to the mixed PKS-NRPS responsible for epothilone biosynthesis, but in fact, a family of docking domains has been identified from this (N-terminus of EpoB) and other such systems, the structure of a model N-terminal docking domain characterized by NMR, and conserved features of the partner C-terminal docking domains (i.e. EpoA referenced in this work) identified (short, negatively-charged region at the extreme C-terminus) (ref. 9) (interestingly, the extreme C-terminus of NatC contains instead a series of positively-charged R). Are the docking domains present at this interface of the same

type? Can the specific engineering strategy be rationalized based on these data? Another important point is how the C-terminal docking domain of NatC was attached to SmlC protein – i.e. what was the fate of the original (true) linker between the SmlC ACP and TE domains? All of this analysis should be clearly presented in order to justify the experimental design (i.e. to support their evolution-inspired design argument, what is the situation in NatC that explains the design of the hybrid SmlC/NatC construct?).

More minor comments:

1. Introduction, page 3: It's not terribly logical to cite colibactin along with the other metabolites, as it is a cause of cancer as opposed to a therapeutic
2. Introduction, page 3: the KSs only catalyze condensation between a starter and an extender unit during the first catalytic cycle – thereafter, between the growing chain and an extender unit; the whole sentence needs to be revised for clarity (the extender unit is thioesterified to an ACP via the action of an acyl transferase...). Same comment for the description of the NRPSs: the amino acid build blocks are activated by the A domains via formation of adenylates, and from there attached to the peptidyl carrier (PCP) domains (as adenylates they are ester derivatives, but once attached to the PCPs, are activated as their thioesters).
3. Introduction, page 3: if the termination domain is a TE, then chain release typically (though not always) occurs by hydrolysis of macrocyclization; if the domain is of another type, then the release reaction may also be different (i.e. the sentence isn't accurate as written)
4. Introduction, page 3: the authors should specify that the work by Bode et al. was carried out with pure NRPS systems, and so the strategy may not necessarily apply to hybrid PKS-NRPS such as those investigated here (there are, for example, differences in quaternary organization, with NRPS being monomeric and hybrid systems likely homodimeric, supporting potential differences in function). The term 'exchange unit' should also be defined for the non-specialist.
5. Introduction, top of page 4: by 'elaborated domain and module organizations', do the authors mean 'elaborate' or 'complex'?
6. It would be useful (i.e. in Fig. 1) to more fully illustrate the biosynthetic pathways to this type of molecule, as the non-expert reader will have difficulty seeing the connection between the organization of the PKS/NRPS and the final product.
7. Bottom of page 4: 'proposed modular structures' should be replaced by 'assembly line organizations' or something analogous, because they are referring to the overall composition in multienzymes of the pathways.
8. Top of page 5: 'reconstructing' is not the correct word, but rather 'rationally modifying' (as the aim was not to reconstruct the native pathways)
9. Results, page 5: what does it mean to 'independently identify' the biosynthetic clusters. If the sequences were already publically available, then the identification was already made. Perhaps they mean that they re-analyzed them to confirm the organization/functional assignments? In the same vein, the sentence (mid-page 5) 'Sequence analyses revealed that...' should be replaced by 'Sequence analysis confirmed that...'
10. Page 5. The authors note that the 'module structures' (rather 'module compositions', as no structural analysis was carried out), were confirmed 'except for that of NatD'. Here, they should explicitly state what difference was noted with the previously-published data.
11. Mid-page 6. The authors refer to 'region 1'. This is in fact a classical ACP four alpha-helix bundle structure. On what basis are the authors suggesting that the intervening loops mediate interactions between the ACP and KS/AT domains, as the cited reference (37) only concerns the KS/ACP interfaces. And what about the newer model for KS/AT/ACP interactions based on the cryo-EM structure of an intact cis-AT PKS module (Dutta, et al. (2014) Nature 510, 512)?
12. Mid-page 6. The presumed C-terminal docking domain of NatC is not a 'loop', nor an 'intermodular linker' (see earlier comment)
13. Top page 7. In fact the authors did not 'replace the AT domain' of SmlC by that of the antimycin system, but instead introduced site-directed mutations in to the SmlC AT domain to confer on it antimycin-type specificity.
14. Page 7 and after. When the authors describe the biosynthesis of new compounds, they should state more systematically the obtained yields (and particularly relative to the parent metabolite),

as this is a long-standing issue in engineering of these compounds – i.e. for the most part, genetic modifications have been accompanied by substantial drops in titre.

15. Mid-page 7. The authors state that the TE domains 'recognize the steric size of the substrate for macrolactonization reactions'. If this is meant to state that TE domains prefer a specific substrate length, this is challenged by some of the literature, particularly that on the TE from the erythromycin PKS which is capable of lactonizing a whole range of different size substrates (6-, 8-, 12-, 14- and 16-membered rings (e.g. Cortes, et al. (1995) *Science* 268, 1487; Kao, et al. (1995) *JACS* 117, 9105; Kao, et al. (1997) *J. Am. Chem. Soc.* 119, 11339–11340; Rowe, et al. (2001) *Chem. Biol.* 8, 475; Martin, et al. (2008) *Org. Biomol. Chem.* 1, 4144). What appears to be crucial instead in this case is the functionality at the acyl terminus and around the attacking nucleophile, as this group presumably binds into a specific recognition pocket adjacent to the acyl terminus. To what TEs are the authors specifically referring?

16. Mid-Page 7. RED-ET should read Red/ET. In the same paragraph, the authors could state that the NMR data were obtained on 'purified 4'. The phrase 'The hydroxyl group at C-1 of 3a...' is confusing, because in fact their referencing the equivalent carbon center in compound 4. The significant figures in the yields need to be corrected (3.9 +/- 0.7 instead of 3.94 +/- 0.66 and 12 +/- 2 (as if the first figure after the . is in doubt, there is no need to quote a second). Finally, although 4 was produced at quite good yield, it was nonetheless 3-fold reduced relative to that of 3a, and so it is not accurate to say that the two yields were 'comparable'.

17. Top of page 8. The structure of 6 should be included in Fig. 4, as this is generated in yields superior to that of the desired compound 5, and also to aid in understanding by the reader. The sentence starting with 'The HMBC correlations...' needs to be corrected (4 should be 5 at the end). Same comment on the significant figures in the errors as above. The explanation for the accumulation of 6 is not entirely clear. What appears to be intended is that the extended residence time of the intermediate on SmlB relative to the native system, leads to its spontaneous hydrolysis. This suggests that transfer between the modified SmlC and NatD is in fact rather inefficient despite their compatible docking domains, leading to a slow-down of all intermediates progressing through the modules. This observation in turn implies that their docking domain engineering may not in fact have been as effective as desired (as is argued on page 10).

18. Bottom page 8. The ccrs are actually crotonyl-CoA reductase/carboxylases, not just reductases. Move 'into the system' after 'we introduced'

19. Top of page 9. For the sentence starting 'At the same time, we also engineered the SmlC PKS module...', suggest completing the sentence with: 'by carrying out modifications in order to broaden its substrate specificity towards that of the antimycin system'. In fact, they don't know that all of the specificity determinants have in fact been transplanted, and so this is a more accurate description of the experiment. Also, they make reference to a motif in an AT domain from the erythromycin PKS which allowed for a change in substrate specificity, but this was only incomplete. Was any of the native compound 2 obtained when AntEV350G was co-expressed, either with the native or modified SmlC AT? (In the discussion (page 11) they appear to state that it is the dominant product, and if this is the case, it should be made clear when the data are first presented). For the sentence 'These data indicated...' suggest 'that the mutations introduced into AT...'

20. Discussion. References 47-50 should be modified to include ref. 12.

21. Discussion. The authors make very brief reference to having also tried a classical strategy for altering AT specificity – AT exchange, but don't provide any experimental details. What sites were tested, and how to these compare to recently-identified 'optimal' fusion junctions (Yuzawa, et al. (2017) *ACS Synth. Biol.* 6, 139)?

22. Discussion. Sentence starting with 'In the complexes with PKSs...' should be revised. Suggestion: 'In the case of systems exhibiting relaxed specificity, this enzyme family could in future be productively used to diversify product structures' (as indeed, the approach wouldn't be limited to pure PKS systems)

23. Discussion. The sentence 'The productivity of the reconstructed modules will be improved by optimizing each structure, based on the structural analysis of each domain or the whole module structure by X-ray crystallography and cryomicroscopy' should be revised. High-resolution structures of each of the domains of PKS and NRPS systems are already available, while analysis

of whole modules remains a technically extremely challenging endeavor (Dutta, et al. (2014) Nature 510, 512) (in fact, no crystal structure of a bonafide modular PKS module has yet been published). Similarly, the final sentence rather overstates the state of play: 'The methodology presented in this study will lead us to efficient module reconstruction to yield super-natural polyketide-nonribosomal peptide antibiotics'. It would be more accurate to say that it confirms that strategies applied to pure PKS and NRPS systems can be productively used with mixed PKS/NRPS.

24. Legend Fig. 1. Suggest: 'Organization of modular enzymes involved...'

25. Legend Fig. 2/Fig. 2: Why was an NRPS module (which presumably contained PCP-TE) used as a template for modeling ACP-TE? To construct this model, were the sequences of the NatC ACP and the SmIC TE artificially joined? Or were the two proteins docked together? Details of how the modeling was carried out should be provided in the Supplementary information. Also, the arrow indicates a 'fusion point' not a 'recombination point'.

26. Legend Fig. 3 and below (there is an odd character overlapping the text). Why are certain atoms in Fig. 3 colored in red (same question for Fig. 4 (part in blue), and Fig. 5 (part in green)). The significant is not obvious because there is no correlation between the color of the modified module/domain and the color of the region of the product affected.

27. Legends: 'The UV chromatogram was monitored at 320 nm', should read 'The chromatogram represents the UV absorbance at 320 nm' or something equivalent.

Reviewer #3 (Remarks to the Author):

The paper by Awakawa et. al uses knowledge of antimycin like NRPS/PKS assembly lines, and bioinformatics insights, to create chimeric NRPS/PKS delivering novel ring expanded, ring contracted and alkyl diversified natural product variants in good yields. The evolutionary similarity of the three NRPS/PKS assembly lines (Ant, SmI & Nat) allows functional fusions to be generated with little loss in productivity. There are few examples of NRPS/PKS engineering working so well, particular in Streptomyces, and therefore this study has sufficient merit for publication. However, there are a number of issues that prevent it being published in its current form, mostly to do with the way the paper is presented:

The paper is not easy to follow and some of the figures are not clear.

The discussion in the introduction on the definition NRPS and PKS (domains) could be abbreviated as this is well known. On the otherhand, the biosynthesis of JBIR-06 and neoantimycin should be explained in more detail to improve overall readability of the manuscript (including the units installed by each module) and where the key differences are between the three pathways.

In addition, the biosynthetic schemes of the three antimycin-like compounds are missing the 3-FSA starter unit PCP (AntG in antimycin biosynthesis) which makes following the biosynthesis from these figures confusing.

It would help understanding of the paper if in addition to labelling the module numbers the unit installed in each case should also be labelled. For example, in figure 1 module 1 installs L-threonine and module 2 incorporates pyruvate, module 3 alkylmalonyl CoA etc... Labelling these fully would also help to easily identify the roles and origins of each module (See Fig. 7 in Nat. Prod. Rep., 2016, 33, 1146-1165 for a much clearer scheme). The colors of each module are also confusing. For example, module 3 in both SmIB and NatB are the same colour, this does help to highlight that these modules are similar insertions relative to AntC, but these modules are not identical as they incorporate different substrates. Modules 2 of AntC, SmIB and NatB also incorporate different substrates but are coloured the same on the diagrams as if they are identical.

On page 5 the authors comment that the bioinformatics analysis revealed a module structure for neoantimycin gene cluster that was the same as that previously reported, but mention that NatD

differs. The authors do not mention what these differences are and why they arise.

Also, on page 5 the manuscript makes mention of Figs 2a and 3a in regards to HPLC analysis of culture extracts of the pKU518J06 and pKU518nant. The correct figures should be 4b and 3b respectively.

On page 9 the authors claim to be first to engineer peptide and polyketide parts for PKS/NRPS products. There have been other examples of PKS/NRPS engineering and it is not necessary to try to claim a first.

Finally, on page 11 the authors state that their method will lead us to.... "super-natural" polyketide-nonribosomal peptide antibiotics. This is an unfortunate wording (hyperbole), are they suggesting they will provide magical antibiotics in the future?

RE: NCOMMS-18-07714-T

"Reprogramming of the antimycin NRPS-PKS assembly lines inspired by gene evolution"

Reviewer #1

1. *a standalone figure for JBIR-06 and neoantimycin production*

According to the suggestion, we added a new supplementary figure (Fig. S3).

2. *Please elaborate what is meant by, "we confirmed that these module structures were the same as those in the previous report, except for NatD".*

Unfortunately, the sequence information from Magarvey's group has not been deposited in the Genbank and not open to public. Therefore, we had to sequence them independently. We explained this in the Introduction (page 5), and newly added a sentence to describe the differences (page 5). "In contrast to the Magarvey's report, where *natD* and *natE* encode A-KR and T-TE domains, respectively, in our sequence data, *natD* encodes a whole single module consisted of C-A-KR-T-TE domains."

3. *The amino acid alignment and assembly line for the ring contraction experiment... are mistaken called out in this sentence: "HPLC analyses of the culture extracts of the recombinants revealed that S. lividans/pKU518J06 and S. lividans/pKU518nant yielded 2 and 3a-3b, respectively (Figs. 2a and 3a)".*

Thank you very much for the notice. We replaced "Figs. 2a and 3a" with "Figs. 3b and 4b".

4. *The supplemental figures containing the NMR and MS analyses establishing the identity of compounds 2, 3a and 3b need to be called out.*

We cited the supplementary Figure S8 and S12-14, as the reviewer suggested.

5. *The paragraph starting 'Notably, in JBIR-06 and neoantimycin biosynthesis...' does not really work well in this section. In my view, it would work better at the start of the AT domain engineering section later on.*

According to the suggestion, we revised the manuscript. "As described above, both of SmIC and NatC PKS modules" (page 9)

6. *The construction of chimeric biosynthetic assembly lines in the supplemental information is a bit difficult to follow. It might be helpful to remove the primers from the text and put all of these in your primer table. It would also be useful to have a table of plasmids constructed and their use in the study.*

According to the suggestion, we newly made tables for the primers/plasmids (Table S1 & S2).

7. *Concerning the sentence, "At the same time, we also engineered the SmIC PKS module by replacing the AT domain...". From what I can tell, the entire domain was not swapped, rather only a few codons were changed. Please amend your language.*

We revised the text as follows. "At the same time, we also engineered SmIC PKS module by introducing mutations into the AT domain in order to broaden its substrate specificity towards that of the antimycin system" (page 9). Please also see our response below (Reviewer #2-19).

8. *(The discussion section) reads more like a summary than an actual discussion section. I appreciate that there is a word limit, but the authors need to more comprehensively put their data into context within the natural products bioengineering field. For instance, aside from the recent paper from Wlodek et al (which the authors do cite), there has been very little if any successful experiments where the 'size' of the chemical scaffold has been rationally engineered. This needs to shine through more.*

We appreciate the thoughtful comments from the reviewer. According to the suggestion, we modified the text as follows. "Considering that there has been very little successful experiments where the size of the chemical scaffold has been rationally engineered, it is quite remarkable that ..." (page 10).

A broader discussion of linker domains and what is known would also seem to be warranted as well. For example, "The sequence of the intermodule linker of NatC was not similar to the one reported in the EpoA study³⁸" is not really meaningful to the reader.

We newly added discussions on the linker and docking domains (page 10, 11).

In addition, during the time this paper was in submission it would seem that structural data for the broad specificity of the antimycin PKS was published in *Angewandte Chemie*. This should be included in the discussion section and the text modified accordingly.

We cited the paper as ref #49.

Reviewer #2

General comment:

Although these experiments were all successful (the spectral data required to validate the structures are included) and the novel compounds obtained in quite respectable yields relative to the parent metabolites, the major criticisms against the work concern both the novelty of the engineering rationale, as well as the specific approaches taken.

We really appreciate the thoughtful and instructive criticism from the reviewer. As the reviewers pointed out, strong point of our manuscript is that our strategy to engineer the three NRPS/PKS assembly lines by taking advantage of bioinformatic analyses and evolutionary insights, successfully generated a set of unnatural novel compounds in practical yields. Indeed, there have been only few examples of NRPS/PKS engineering working so well. The construction of the system has not been accomplished without the guidance of the bioinformatic analyses, and the facile genetic manipulation by virtue of the *E. coli-Streptomyces* shuttle BAC vector. The importance of this study is to demonstrate the utility of the strategy, and serve the knowledge to accomplish a general algorithm for PKS-NRPS module reconstitution. The following sentences were added to emphasize our motivation in the introduction section (page 4).

“The knowledge on freedom and constraint for NRPS-PKS engineering should be accumulated more to understand how this system has been evolved so far and can be artificially evolved in the future. This would lead to construction of algorithm such as the computational platforms which clearly predict domain/linker/docking-domain boundaries in *cis*-PKS system for designing chimeric modules¹⁶”

Further, the experiment in which the AntE (CCR) is combined with mutagenesis of the SmIC AT domain is effective, and the ring-expansion of JBIR-06 *via* docking domain engineering is notable, especially in the aspect of the yield. Finally, we believe that our results should be beneficial to furnish the knowledge on future engineering studies and develop the informatics tools for module enzymes. To emphasize these points, we added following sentences in the Discussion (page 10-12).

“Considering that there has been very little successful experiments where the size of the chemical scaffold has been rationally engineered, it is quite remarkable that the yields of the compounds which we obtained in the engineered NRPS-PKS system are 5-10 times higher than those in the reported module assembly line engineering^{11,46-48}.” “This accomplishment was done with guidance of bioinformatic analysis of the co-evolved module structures.” “Furthermore, it furnishes the knowledge on the module engineering studies, and helps to construct the new algorithm to yield unnatural polyketide-nonribosomal peptide assembly lines.”

Major comments #1

*Revisiting the ‘evolutionary’ justification for the modifications made, several other issues need to be considered...In any case, the specific evolutionary history of hybrid *cis*-AT PKS/NRPS has yet to be rigorously investigated. In this context, the description of how PKSs have evolved (top of page 4) is too simplistic.*

According to the suggestion, we revised the text as follows. “Thus, for *cis*-AT PKSs, the predominant evolution model is repeated duplication of ancestral modules coupled with domain exchanges^{12-13, 15}, while for the distinct class of *trans*-AT PKSs, it is rather horizontal gene transfer that appears to dominate¹⁴⁻¹⁵. However, the specific evolutionary history of hybrid *cis*-AT PKS/NRPS system has yet to be rigorously investigated.” (page 4)

*The authors then make specific reference (on page 4) to a revised model of evolution of *cis*-AT PKSs...*

We deleted ref #18 to avoid misunderstanding, as we agree that it is only a specific example as the reviewer suggested.

The sentence starting ‘This is a promising way to produce...’ (top of page 4) is also problematic for a number of reasons, most notably because this paper doesn’t address the ‘reconstructing of domain structures’, ...

According to the suggestion, we revised the text as follows. “It is a promising way to rationally modify the module enzymes by reconstructing the modules according to the evolutionary course

in nature, because this method is likely to maintain the connectivity between modules with minimizing the change of the structure, as well as the protein-protein interactions which underlie their function.” (page 4)

Finally, the bioinformatics analysis described on page 5 runs counter to an analysis of the same PKSs published by others..... However, Magarvey et al. specifically argue based on their independent phylogenetic analysis, that neither of these events is likely. The discrepancy between these analyses needs to be explained, and the phylogenetic data supporting the evolutionary model proposed here need to be presented.

We appreciate the thoughtful comment again. We compared the DNA sequence of adenylation domains (please see the newly presented Fig. S4), and found that our statement was wrong. We revised the text. “Sequence alignment of the A domains from *ant*, *sml*, and *nat* system suggested that the module 1 is derived from gene duplication of the same ancestor, while the second, third, and fifth modules likely evolved from different sources (Fig. S4).” (page 6)

Major comments #2

The authors need to be more rigorous in their use of PKS/NRPS-specific vocabulary. Sequences within PKS or NRPS polypeptides (i.e. intrasubunit) that connect functional domains and modules are referred to as ‘linkers’, while those that mediate communication between subsequent subunits (i.e. intersubunit) are referred to as ‘docking domains’ (see refs. 7, 9 and 10).

According to the suggestion, we revised the text.

Are the docking domains present at this interface of the same type? Can the specific engineering strategy be rationalized based on these data? Another important point is how the C-terminal docking domain of NatC was attached to SmlC protein – i.e. what was the fate of the original (true) linker between the SmlC ACP and TE domains? All of this analysis should be clearly presented in order to justify the experimental design (i.e. to support their evolution-inspired design argument, what is the situation in NatC that explains the design of the hybrid SmlC/NatC construct?).

We appreciate the thoughtful comments again. Our sequence analysis revealed that the C-terminal region of NatC contains a conserved basic acidic residue, which likely plays important role in the domain interactions, as shown in Richter et al. *Nat. Chem. Biol.* 2008, 4, 75. Further, we also found that NatD does not maintain the N-terminal secondary structure annotated as a docking domain of EpoB in Dowling et al. *PNAS* 2016, 113, 12432. This finding suggests that NatD can interact with NatC or SmlC in different manner from EpoB. Please see newly added Figure S6A and newly added explanations as follows. “Notably, our sequence analysis of the C-terminal region of NatC revealed that while it is not highly similar to the docking domains of the other PKS modules that interact with NRPS, NatC₁₃₉₁₋₁₄₁₁ contains a conserved basic residue (Arg₁₃₉₆) (Fig. S6A), which may play an important role in interaction with the NRPS module.⁸ Further, the three α -helices and two β -turns, annotated as a docking domain in the EpoB NRPS structure⁹, are uniquely replaced with a shorter 9 amino acid sequences in NatD (Fig. S6B). This suggests the possibility that NatD can interact with NatC or SmlC in significantly different manner from that of EpoB.” (page 7)

Minor comments

1. Introduction, page 3: It’s not terribly logical to cite colibactin along with the other metabolites, as it is a cause of cancer as opposed to a therapeutic

Thank you for the suggestion. We removed the citation of colibactin.

2. Introduction, page 3: the KSs only catalyze condensation between a starter and an extender unit during the first catalytic cycle – thereafter, between the growing chain and an extender unit; the whole sentence needs to be revised for clarity (the extender unit is thioesterified to an ACP via the action of an acyl transferase...). Same comment for the description of the NRPSs: the amino acid build blocks are activated by the A domains via formation of adenylates, and from there attached to the peptidyl carrier (PCP) domains (as adenylates they are ester derivatives, but once attached to the PCPs, are activated as their thioesters).

According to the suggestion, we revised the text. (page 3)

3. Introduction, page 3: if the termination domain is a TE, then chain release typically (though not always) occurs by hydrolysis of macrocyclization; if the domain is of another type, then the release reaction may also be different

According to the suggestion, we modified the text as follows. “In both systems, if the termination domain is a thioesterase (TE), then chain release typically occurs by hydrolysis or macrocyclization.” (page 3)

4. Introduction, page 3: the authors should specify that the work by Bode et al. was carried out with pure NRPS systems, and so the strategy may not necessarily apply to hybrid PKS-NRPS such as those investigated here. The term 'exchange unit' should also be defined for the non-specialist.

We appreciate the thoughtful suggestion. We revised the text as follows. "... the exchange unit that is a set of A-T-C domains transplantable into the module¹¹. However, this strategy may not necessarily apply to hybrid NRPS-PKS systems because the quaternary organization of NRPS-PKS, consisted of monomeric NRPS and dimeric PKS, is likely to be different from that of pure NRPS system." (page 3)

5. Introduction, top of page 4: by 'elaborated domain and module organizations', do the authors mean 'elaborate' or 'complex'?

We apologize for any confusion. We deleted the phrase to avoid misleading.

6. It would be useful (i.e. in Fig. 1) to more fully illustrate the biosynthetic pathways to this type of molecule, as the non-expert reader will have difficulty seeing the connection between the organization of the PKS/NRPS and the final product.

According to the suggestion, we revised Figure 1 and added more detailed explanation in the legends. Please also see our response below (Reviewer #3-3).

7. Bottom of page 4: 'proposed modular structures' should be replaced by 'assembly line organizations' or something analogous, because they are referring to the overall composition in multienzymes of the pathways.

We replaced 'proposed modular structures' with 'assembly line organizations'. (page 5)

8. Top of page 5: 'reconstructing' is not the correct word, but rather 'rationally modifying' (as the aim was not to reconstruct the native pathways)

According to the suggestion, we revised the text.

9. Results, page 5: what does it mean to 'independently identify' the biosynthetic clusters. If the sequences were already publically available, then the identification was already made. Perhaps they mean that they re-analyzed them to confirm the organization/functional assignments? In the same vein, the sentence (mid-page 5) 'Sequence analyses revealed that...' should be replaced by 'Sequence analysis confirmed that...'

Unfortunately, the sequence information from Magarvey's group has not been deposited in the Genbank and not open to public. Therefore, we sequenced them independently. We explained this in the Introduction. Please also see our response above (Reviewer #1-2).

10. Page 5. The authors note that the 'module structures' (rather 'module compositions', as no structural analysis was carried out), were confirmed 'except for that of NatD'. Here, they should explicitly state what difference was noted with the previously-published data.

According to the suggestion, we revised the text. Please see our reply above (Reviewer #1-2).

11. Mid-page 6. The authors refer to 'region 1'. This is in fact a classical ACP four alpha-helix bundle structure. On what basis are the authors suggesting that the intervening loops mediate interactions between the ACP and KS/AT domains, as the cited reference (37) only concerns the KS/ACP interfaces. And what about the newer model for KS/AT/ACP interactions based on the cryo-EM structure of an intact cis-AT PKS module (Dutta, et al. (2014) Nature 510, 512)?

Thank you for the instructive comment. Yes, as the reviewer pointed out, we defined the ACP region, based on the classical ACP structure. The interactions of the loop and helix of ACP was indicated in ref #35, which was further clarified with the cryo-EM study by Dutta. We now cited the Nature paper as ref #36. (page 6)

12. Mid-page 6. The presumed C-terminal docking domain of NatC is not a 'loop', nor an 'intermodular linker' (see earlier comment)

We corrected the text with "docking domain".

13. Top page 7. In fact the authors did not 'replace the AT domain' of SmlC by that of the antimycin system, but instead introduced site-directed mutations in to the SmlC AT domain to confer on it antimycin-type specificity.

According to the suggestion, we revised the text. Please also see our response above (Reviewer #1-7).

14. Page 7 and after. When the authors describe the biosynthesis of new compounds, they should state more systematically the obtained yields (and particularly relative to the parent

metabolite), as this is a long-standing issue in engineering of these compounds – i.e. for the most part, genetic modifications have been accompanied by substantial drops in titre.

Thank you for the thoughtful suggestion. We modified the text to emphasize the obtained yields, and newly added Table 1 to summarize the results.

15. Mid-page 7. The authors state that the TE domains ‘recognize the steric size of the substrate for macrolactonization reactions’. If this is meant to state that TE domains prefer a specific substrate length, this is challenged by some of the literature, particularly that on the TE from the erythromycin PKS which is capable of lactonizing a whole range of different size substrates (6-, 8-, 12-, 14- and 16-membered rings (e.g. Cortes, et al. (1995) *Science* 268, 1487; Kao, et al. (1995) *JACS* 117, 9105; Kao, et al. (1997) *J. Am. Chem. Soc.* 119, 11339–11340; Rowe, et al. (2001) *Chem. Biol.* 8, 475; Martin, et al. (2008) *Org. Biomol. Chem.* 1, 4144). What appears to be crucial instead in this case is the functionality at the acyl terminus and around the attacking nucleophile, as this group presumably binds into a specific recognition pocket adjacent to the acyl terminus. To what TEs are the authors specifically referring?

We appreciate the thoughtful comment again. According to the suggestion, we revised the text as follows. “TE domains usually exhibit tolerant substrate specificities, as exemplified by early studies where a TE domain from erythromycin is tolerant for accepting diverse-length of polyketide chain³⁸, and recognize the functionality at the acyl terminus and around the attacking nucleophile³⁹. Therefore, we expected that TE_{SmlC} domain, which recognize the acyl group derived from polyketide and the hydroxyl group of threonine as a nucleophile, accepts the intermediate from the new construct and catalyzes macrocyclization.” (page 7)

16. Mid-Page 7. RED-ET should read Red/ET.

We revised the text as suggested.

In the same paragraph, the authors could state that the NMR data were obtained on ‘purified 4’. The phrase ‘The hydroxyl group at C-1 of 3a...’ is confusing, because in fact their referencing the equivalent carbon center in compound 4. The significant figures in the yields need to be corrected (3.9 +/- 0.7 instead of 3.94 +/- 0.66 and 12 +/- 2 (as if the first figure after the . is in doubt, there is no need to quote a second). Finally, although 4 was produced at quite good yield, it was nonetheless 3-fold reduced relative to that of 3a, and so it is not accurate to say that the two yields were ‘comparable’.

According to the suggestion, we revised the text.

17. Top of page 8. The structure of 6 should be included in Fig. 4, as this is generated in yields superior to that of the desired compound 5, and also to aid in understanding by the reader.

We modified Fig.4 as suggested.

The sentence starting with ‘The HMBC correlations...’ needs to be corrected (4 should be 5 at the end). Same comment on the significant figures in the errors as above.

According to the suggestion, we revised the text.

The explanation for the accumulation of 6 is not entirely clear. What appears to be intended is that the extended residence time of the intermediate on SmlB relative to the native system, leads to its spontaneous hydrolysis. This suggests that transfer between the modified SmlC and NatD is in fact rather inefficient despite their compatible docking domains, leading to a slow-down of all intermediates progressing through the modules. This observation in turn implies that their docking domain engineering may not in fact have been as effective as desired (as is argued on page 10).

As suggested, we revised the text as follows. “The unexpected accumulation of the acyclic 6 (9.7 ± 0.8 mg/L) might have been caused by the extended residence time of the intermediate on SmlB relative to the native system, leading to its spontaneous hydrolysis.” (page 8)

18. Bottom page 8. The ccrs are actually crotonyl-CoA reductase/carboxylases, not just reductases. Move ‘into the system’ after ‘we introduced’

According to the suggestion, we revised the text.

19. Top of page 9. For the sentence starting ‘At the same time, we also engineered the SmlC PKS module...’, suggest completing the sentence with: ‘by carrying out modifications in order to broaden its substrate specificity towards that of the antimycin system’. In fact, they don’t know that all of the specificity determinants have in fact been transplanted, and so this is a more accurate description of the experiment. Also, they make reference to a motif in an AT domain from the erythromycin PKS which allowed for a change in substrate specificity, but this was only incomplete.

According to the suggestion, we revised the text. Please also see our response above (Reviewer #1-7). “At the same time, we also engineered SmlC PKS module by introducing mutations into the AT domain in order to broaden its substrate specificity towards that of the antimycin system”, “We focused on this motif since it has been reported that the mutation of this site “VDYASH” in the AT domain of the erythromycin PKS altered the substrate preference⁴²⁻⁴⁵.” (page 9)

Was any of the native compound 2 obtained when AntEV350G was co-expressed, either with the native or modified SmlC AT? (In the discussion (page 11) they appear to state that it is the dominant product, and if this is the case, it should be made clear when the data are first presented).

To make it clear, we added explanations as follows. “However, it should be noted that the yield of the native product **2** is still dominant (5.0 ± 1.7 mg/L) (Table 1), indicating further optimization is required. Nonetheless, it was remarkable that the total yield of the newly obtained JBIR-06 derivatives with different alkyl chain structures, **7a**, **7c**, **7e** and **7f** (3.8 mg/L), is almost comparable with that of **2**.” (page 10)

For the sentence ‘These data indicated...’ suggest ‘that the mutations introduced into AT...’
We modified the text, as suggested.

20. Discussion. References 47-50 should be modified to include ref. 12.
We modified the references, as suggested.

21. Discussion. The authors make very brief reference to having also tried a classical strategy for altering AT specificity – AT exchange, but don’t provide any experimental details. What sites were tested, and how to these compare to recently-identified ‘optimal’ fusion junctions (Yuzawa, et al. (2017) ACS Synth. Biol. 6, 139)?

According to the suggestion, we now provided details for the AT swapping experiments which we tried in the text and Fig. S11, and compared it with the optimal fusion junctions that Yuzawa et al. suggested (this information is now cited as ref #50). “With expecting the higher yield, we also tested several chimeric PKS module constructs such as $KS_{SmlC}-AT_{AntD}-MT\ ACP\ TE_{SmlC}$ with domain swapping strategy (Figure S11), however none of them afforded any detectable products (data not shown). Further adoption of the recently-identified ‘optimal’ fusion junctions for AT-domain swapping⁵⁰ may improve the product yields.” (page 11)

22. Discussion. Sentence starting with ‘In the complexes with PKSs...’ should be revised. Suggestion: ‘In the case of systems exhibiting relaxed specificity, this enzyme family could in future be productively used to diversify product structures’ (as indeed, the approach wouldn’t be limited to pure PKS systems)

We revised the text, as suggested.

23. Discussion. The sentence ‘The productivity of the reconstructed modules will be improved by optimizing each structure, based on the structural analysis of each domain or the whole module structure by X-ray crystallography and cryomicroscopy’ should be revised. High-resolution structures of each of the domains of PKS and NRPS systems are already available, while analysis of whole modules remains a technically extremely challenging endeavor (Dutta, et al. (2014) Nature 510, 512) (in fact, no crystal structure of a bonafide modular PKS module has yet been published).

According to the suggestion, we revised the text as follows. “The productivity of the engineered modules would be improved by optimizing the interactions between domains and modules based on the structural analysis of the whole module complex, even though it still remains a technically extremely challenging endeavor.” (page 12)

Similarly, the final sentence rather overstates the state of play: ‘The methodology presented in this study will lead us to efficient module reconstruction to yield super-natural polyketide-nonribosomal peptide antibiotics’. It would be more accurate to say that it confirms that strategies applied to pure PKS and NRPS systems can be productively used with mixed PKS/NRPS.

We modified the text as suggested.

24. Legend Fig. 1. Suggest: ‘Organization of modular enzymes involved...’
According to the suggestion, we revised the legend.

25. Legend Fig. 2/Fig. 2: Why was an NRPS module (which presumably contained PCP-TE) used as a template for modeling ACP-TE? Details of how the modeling was carried out should

be provided in the Supplementary information. Also, the arrow indicates a 'fusion point' not a 'recombination point'.

We apologize for any confusion. According to the suggestion, we added the information for the homology modeling in the Supporting information (page S5), and modified Fig. 2.

To construct this model, were the sequences of the NatC ACP and the SmlC TE artificially joined? Or were the two proteins docked together?

The former is correct. Please see the details in the Supporting information (page S8).

26. Legend Fig. 3 and below (there is an odd character overlapping the text). Why are certain atoms in Fig. 3 colored in red (same question for Fig. 4 (part in blue), and Fig. 5 (part in green)). The significant is not obvious because there is no correlation between the color of the modified module/domain and the color of the region of the product affected.

According to the suggestion, we changed the color in the Figs. 3-5.

27. Legends: 'The UV chromatogram was monitored at 320 nm', should read 'The chromatogram represents the UV absorbance at 320 nm' or something equivalent.

According to the suggestion, we revised the legends.

Reviewer #3

1. The discussion in the introduction on the definition NRPS and PKS (domains) could be abbreviated as this is well known. On the other hand, the biosynthesis of JBIR-06 and neoantimycin should be explained in more detail to improve overall readability of the manuscript (including the units installed by each module) and where the key differences are between the three pathways.

We appreciate the thoughtful suggestion. We revised the text and explained the biosynthesis of JBIR-06 and neoantimycin in more detail in introduction (Page 5) and in the legend of Fig. 1. Please also see our response above (Reviewer #2-6).

2. In addition, the biosynthetic schemes of the three antimycin-like compounds are missing the 3-FSA starter unit PCP (AntG in antimycin biosynthesis) which makes following the biosynthesis from these figures confusing.

According to the suggestion, we revised Figs. 1 & 3-5.

3. It would help understanding of the paper if in addition to labelling the module numbers the unit installed in each case should also be labelled. For example, in figure 1 module 1 installs L-threonine and module 2 incorporates pyruvate, module 3 alkylmalonyl CoA etc... Labelling these fully would also help to easily identify the roles and origins of each module (See Fig. 7 in Nat. Prod. Rep., 2016, 33, 1146-1165 for a much clearer scheme). The colors of each module are also confusing. For example, module 3 in both SmlB and NatB are the same colour, this does help to highlight that these modules are similar insertions relative to AntC, but these modules are not identical as they incorporate different substrates. Modules 2 of AntC, SmlB and NatB also incorporate different substrates but are coloured the same on the diagrams as if they are identical.

According to the suggestion, we modified Figs. 1 & 3-5. Please also see our response above (Reviewer #2-6).

4. On page 5 the authors comment that the bioinformatics analysis revealed a module structure for neoantimycin gene cluster that was the same as that previously reported, but mention that NatD differs. The authors do not mention what these differences are and why they arise.

We revised the text. Please see our responses above (Reviewer #1-2 and Reviewer #2-9).

5. Also, on page 5 the manuscript makes mention of Figs 2a and 3a in regards to HPLC analysis of culture extracts of the pKU518J06 and pKU518nant. The correct figures should be 4b and 3b respectively.

We revised it. Thank you.

6. On page 9 the authors claim to be first to engineer peptide and polyketide parts for PKS/NRPS products. There have been other examples of PKS/NRPS engineering and it is not necessary to try to claim a first.

According to the suggestion, we deleted the sentence.

7. Finally, on page 11 the authors state that their method will lead us to.... "super-natural" polyketide-nonribosomal peptide antibiotics. This is an unfortunate wording (hyperbole), are they suggesting they will provide magical antibiotics in the future?

Yes, we agree with the reviewer. We removed the sentence.

Another minor point. We deleted some misplaced spectra from Fig. S8.

We hope you will agree that the manuscript has been significantly improved, and that you will find it acceptable for publication.

Reviewers' comments:

Reviewer #1 (Remarks to the Author):

This is to confirm that initial comments/revisions from Reviewer 1 have been satisfactorily addressed in the revised version of the manuscript.

However, during the review/revision process, an article describing the identification and characterization of the neoantimycin biosynthetic gene cluster was published (<https://www.ncbi.nlm.nih.gov/pubmed/29693372>). This article should be cited in the introduction and in the first section of the results. The article alleviates the problem the seemingly incomplete work from the Magarvey group regarding the composition of the neoantimycin gene cluster, i.e. this reference should supersede reference 34.

I have made several changes to the revised text, which can be seen in the attached annotated PDF.

Reviewer #2 (Remarks to the Author):

The authors have carefully addressed the majority of my comments, making the ms ultimately suitable for publication in Nat. Commun.

A couple further modifications are, however, required:

The overall ms could productively be edited for English language usage, as there are some instances where the meaning is obscured.

Page 3, lines 58-59: the phrase 'linker domain between two modules is referred to as a docking domain', is not correct. The word linker refers to amino acid sequences WITHIN subunits that link modules, while docking domains mediate communication between modules located on DISTINCT subunits.

Page 3, lines 70-72. In fact, it cannot be assumed that NRPSs within the contexts of hybrid PKS-NRPS are in fact monomeric (as for pure NRPSs). For example, the docking domain from an NRPS subunit (TubC) within the hybrid tubulysin PKS-NRPS was found to be homodimeric, suggesting that NRPSs in this context may even be homodimeric. The authors could also reference this article more explicitly (ref. 8) as in fact, this structure preceded that of the EpoB docking domain.

Page 6, lines 135-137: the sentence 'DNA sequence alignment of the A domains from ant, sml, and nat system suggested that the module 1 is derived from gene duplication of the same ancestor, while the second, third, and fifth modules likely evolved from different sources (Supplementary Fig. 4)' doesn't make sense, as 'gene duplication' implies that there are multiple modules in EACH of the three systems that are derived from the same ancestral module. In fact, what they find is that homologous module 1s are present in the three assembly lines, no? Also, what are the relationships between the other modules between the three systems – they should precisely state which module/subunit of each system corresponds to the other, as this formed the basis for their sequence comparisons (in this context, an additional figure in the supplementary section would be quite useful).

Page 7, lines 154: homology modeling cannot 'confirm' that a sequence region adopts a particular structure, as actual structural data are required – but it can support the hypothesis. Line 155: a 'docking domain' is not a 'loop'

Page 7, lines 159-161. The authors mention that the C-terminal region of NatC contains A conserved Arg. What is it conserved with, as the alignment doesn't indicate conservation? In fact, while other C-terminal docking domains that interact with NRPS subunits contain a run of negatively-charged residues, the C-terminus of NatC contains SEVERAL positively-charged residues, which might play a role in determining interaction specificity, and are certainly consistent with the idea that docking between NatC/NatD is fundamentally different than for other interfaces.

The authors could point out here that despite these dissimilarities, their engineering strategy worked.

Page 8, lines 192-194. The authors should mention, as requested in my previous review, the fate of the original (true) linker between the SmIC ACP and TE domains. Was it preserved intact, and the docking domain attached to its C-terminal end, for example?

Reviewer #3 (Remarks to the Author):

“1. The discussion in the introduction on the definition of NRPS and PKS (domains) could be abbreviated as this is well known. On the other hand, the biosynthesis of JBIR-06 and neoantimycin should be explained in more detail to improve overall readability of the manuscript (including the units installed by each module) and where the key differences are between the three pathways.
2. In addition, the biosynthetic schemes of the three antimycin-like compounds are missing the 3-FSA starter unit PCP (AntG in antimycin biosynthesis) which makes following the biosynthesis from these figures confusing.
3. It would help the understanding of the paper if in addition to labelling the module numbers the unit installed in each case could also be labelled. For example, in figure 1 module 1 installs Lthreonine and module 2 incorporates pyruvate, module 3 alkylmalonyl CoA etc... Labelling these fully would also help to easily identify the roles and origins of each module (See Fig. 7 in Nat. Prod. Rep., 2016, 33, 1146-1165 for a much clearer scheme).”

These comments have been fully addressed by the new additions made to the manuscript. The route through the biosynthesis is now much more easily followed.

“The colors of each module are also confusing. For example, module 3 in both SmlB and NatB are the same colour, this does help to highlight that these modules are similar insertions relative to AntC, but these modules are not identical as they incorporate different substrates. Modules 2 of AntC, SmlB and NatB also incorporate different substrates but are coloured the same on the diagrams as if they are identical.”

The authors have altered the manuscript to make it clearer that apart from the starter module and the first module of the NRPS, the other modules are more distantly related. However in the Figure 1 legend the authors are still claiming that module 2 is conserved throughout the three systems whereas in reality the three modules incorporate different units (pyruvate, isoleucic acid, valid acid) and SmlB_A2 only shows 34% similarity to AntC_A2 of, while in fact SmlB_A3 which the authors claim is a relative insertion shows higher similarity (61%) to AntC_A2 suggesting if any module has been inserted that module 2 is the more likely candidate. The insertion of module 3 in neoantimycin relative to antimycin (NatB) is not discussed at all in the figure 1 legend.

Page 6, line 136 more accurately describes the relationships between the clusters by only suggesting that only module 1 are conserved “derived from gene duplication of the same ancestor”. The figure legend and the text should be brought into agreement with each other. Additionally Supplementary figure 4, which displays the percentage similarity between AntC, SmlB and NatB is hard to read. The modules with highest similarity and therefore predicted to have an ancestor in common (e.g. Greater than 79% similarity in this figure) should be highlighted to make these related modules more obvious and the modules that are closely related (maybe greater than 61% in this figure) highlighted in a different shading/colour. Since the authors are suggesting that they are following an evolutionary approach to hybrid NRPS/PKS engineering then these evolutionary relationships/differences should be clearer.

“4. On page 5 the authors comment that the bioinformatics analysis revealed a module structure for neoantimycin gene cluster that was the same as that previously reported, but mention that NatD differs. The authors do not mention what these differences are and why they arise.”

The authors have now detailed what the differences are between the NatD they sequenced for this study and the example found in the literature and explain that the previous annotation is not available for direct comparison (Page 5 line 123). However it seems that there are more variations between the two clusters other than NatD. Comparison with the previously published annotation reveals that NatE also varies between the two studies - the identically named genes are located in

opposite reading frames and have dissimilar functions (MbtH in this study and PKS in the previous). The direction and size of NatG-NatR are also in disagreement with different assigned functions (based on BLAST analysis) in each study. This potentially suggests that the two sequences may originate from different organisms and/or slightly different clusters, or may just arise from differences in sequencing quality between the two studies. Without having access to the data from the other study it is impossible to say why these disparities occur but although these do not ultimately change the outcome of the work they should not be hidden and perhaps a sentence detailing the exact differences should be included for completions sake (or a supplementary figure).

“5. Also, on page 5 the manuscript makes mention of Figs 2a and 3a in regards to HPLC analysis of culture extracts of the pKU518J06 and pKU518nant. The correct figures should be 4b and 3b respectively.

6. On page 9 the authors claim to be first to engineer peptide and polyketide parts for PKS/NRPS products. There have been other examples of PKS/NRPS engineering and it is not necessary to try to claim a first.

7. Finally, on page 11 the authors state that their method will lead us to.... "super-natural" polyketide-nonribosomal peptide antibiotics. This is an unfortunate wording (hyperbole), are they suggesting they will provide magical antibiotics in the future?”

These comments have all been addressed.

Additional comments:

- Page 8 line 186 – The authors attribute the lack of KR (NatF) activity with the ring contracted neoantimycin to polar effects on expression of NatF. What is the evidence for this? Perhaps the enzyme no longer recognises the substrate? What is the sequence similarity to AntM? Do these discrete KR domains often show relaxed substrate specificity?
- The term ACP is used in figure 1 and the text while T domain is used in figures 3, 4 and 5. This should ideally be consistent throughout the manuscript.
- I suggest changing the first sentence in the introduction to ‘are an important class of natural products’ rather than major resources as written.
- The discussion section contains a lot of repetition from the results section. I would recommend limiting the discussion to some simple concluding remarks and outlooks such as those starting on page 12 line 292.

General Comments:

In general the additions to the paper do improve the quality of the manuscript. However the added text contains multiple errors and needs to be proof read more extensively before the paper is appropriate for publication. Some examples are as follows:

- Page 3 line 59 – The sentence ‘which defines the order of chain transfer to avoid unwanted transfer’ should be reworded so the meaning is clearer.
- Page 3 line 65 refers to ‘these engineering studies’ but no engineering studies seem to have been cited prior to that claim.

- Page 3 line 69 should be changed to something like 'using exchange units that are sets of A-T-C domains that can be transplanted into active chimeric modules'.
- Page 4 line 83 change 'with' to 'while'
- Page 4 line 86 consider rewording to 'could lead to computational platforms which clearly predict...'
- On Page 6, to improve the flow of the text, the paragraph beginning line 150... 'As a result' to Page 7 line 157 'NRPS-PKS system' which details the bioinformatic annotation should be moved up to line 142 and go before the sentence starting 'therefore with the bioinformatics information'.
- Page 7 – line 170 – 175. This paragraph should be reworded so that its meaning is clearer.
- Page 8 line 185 – consider changing 'reduction' to 'loss' as reduction is used in its chemical sense elsewhere in the paper.

Reviewer #1

1. *However, during the review/revision process, an article describing the identification and characterization of the neoantimycin biosynthetic gene cluster was published (<https://www.ncbi.nlm.nih.gov/pubmed/29693372>). This article should be cited in the introduction and in the first section of the results. The article alleviates the problem the seemingly incomplete work from the Magarvey group regarding the composition of the neoantimycin gene cluster, i.e. this reference should supersede reference 34.*

Thank you very much for the information. According to the suggestion, we cited the paper as ref #32 in the introduction and in the first section of the results.

“Very recently, Zhang’s group also reported the neoantimycin gene cluster ...” (page 5)

“As Zhang’s group also reported the same NatD domain organization ...” (page 5)

2. *I have made several changes to the revised text, which can be seen in the attached annotated PDF.*

According to the reviewer’s thoughtful suggestion, we revised the text as follows.

“and thioesterified to the thiolation (T) domain ...” (Page 3)

“the predominant evolutionary model is a repeated cycle of ...” (Page 4)

“It is a promising way to modify the module compositions ...” (Page 4)

“the TE domain from erythromycin system was shown to accept ...” (Page 7)

“the Rec/ET system” (Page 7)

“because the N-terminus of NatD lacks a predicted secondary structure ...” (Page 11)

We deleted the sentence “Such structural information also benefits ... to produce 5.” (Page 11)

Reviewer #2

1. *Page 3, lines 58-59: the phrase ‘linker domain between two modules is referred to as a docking domain’, is not correct. The word linker refers to amino acid sequences WITHIN subunits that link modules, while docking domains mediate communication between modules located on DISTINCT subunits.*

We really appreciate the reviewer’s thoughtful and constructive comments. According to the suggestion, we revised the text as follows.

“The amino acid sequences within subunits is referred to as a linker domain, and those which are located on distinct subunits and mediate communication between modules are referred to as a docking domain⁸⁻⁹.” (page 3)

2. Page 3, lines 70-72. *In fact, it cannot be assumed that NRPSs within the contexts of hybrid PKS-NRPS are in fact monomeric (as for pure NRPSs). For example, the docking domain from an NRPS subunit (TubC) within the hybrid tubulysin PKS-NRPS was found to be homodimeric, suggesting that NRPSs in this context may even be homodimeric. The authors could also reference this article more explicitly (ref. 8) as in fact, this structure preceded that of the EpoB docking domain.*

We revised the text, and newly cited the ref #12, according to the reviewer's suggestion.

“However, this strategy may not be necessarily applied to hybrid NRPS-PKS systems, because the quaternary organization of NRPS-PKS is likely different from that of pure NRPS system. In fact, the quaternary structure of NRPS module is defined as a monomer in the early chromatographic study¹², but we cannot exclude that it works as a dimer in the NRPS-PKS system as the docking domain from tubulysin system forms a homodimer in the solution⁹.” (page 3-4) “The interactions between PKS and NRPS modules are still elusive, and need to be clarified by X-ray crystallization or cryo-EM analyses for future engineering studies.” (page 11)

3. Page 6, lines 135-137: *the sentence ‘DNA sequence alignment of the A domains from ant, sml, and nat system suggested that the module 1 is derived from gene duplication of the same ancestor, while the second, third, and fifth modules likely evolved from different sources (Supplementary Fig. 4)’ doesn’t make sense, as ‘gene duplication’ implies that there are multiple modules in EACH of the three systems that are derived from the same ancestral module. In fact, what they find is that homologous module 1s are present in the three assembly lines, no? Also, what are the relationships between the other modules between the three systems – they should precisely state which module/subunit of each system corresponds to the other, as this formed the basis for their sequence comparisons (in this context, an additional figure in the supplementary section would be quite useful).*

According to the suggestion, we modified the text as follows. “DNA sequence alignment of the A domains from *ant*, *sml*, and *nat* system suggested that the first modules in each system are highly similar each other, implying that they might be derived from the same ancestor.” (page 6)

Further, we added a new Figure S4B for the sequence comparisons. Please also see our responses below (Reviewer #3-1 and 3-2).

4. Page 7, lines 154: *homology modeling cannot ‘confirm’ that a sequence region adopts a particular structure, as actual structural data are required – but it can support the hypothesis. Line 155: a ‘docking domain’ is not a ‘loop’*

According to the suggestion, we revised the text. “Homology modeling (SWISS-MODEL server, <https://swissmodel.expasy.org>) supports the hypothesis on its structure including region 1 forms four α -helices and two loops, which are involved in the interactions with the KS-AT domains³⁷⁻³⁸.” (page 6)

5. Page 7, lines 159-161. *The authors mention that the C-terminal region of NatC contains A conserved Arg. What is it conserved with, as the alignment doesn’t indicate conservation? In fact, while other C-terminal docking domains that interact with NRPS subunits contain a run of negatively-charged residues, the C-terminus of NatC contains SEVERAL positively-charged residues, which might play a role in determining interaction specificity, and are certainly consistent with the idea that*

docking between NatC/NatD is fundamentally different than for other interfaces.

The authors could point out here that despite these dissimilarities, their engineering strategy worked.

Thank you very much for the thoughtful suggestion. We revised the text as follows.

“Notably, our sequence analysis of the C-terminal region of NatC revealed that NatC₁₃₉₁₋₁₄₁₁ contains several positively-charged residues while other C-terminal docking domains that interact with NRPS subunits contain a run of negatively-charged residues.” (page 7)

“This information suggests that the docking between NatC and NatD is fundamentally different from other interfaces. Despite these dissimilarities, our engineering strategy with the docking domain of NatC worked (see the section of ring expansion).” (page 7)

6. Page 8, lines 192-194. The authors should mention, as requested in my previous review, the fate of the original (true) linker between the SmlC ACP and TE domains. Was it preserved intact, and the docking domain attached to its C-terminal end, for example?

Thank you for the suggestion. The linker and TE was replaced with the docking domain at the same time. We modified the text as follows. “To do so, we appended the NatD NRPS module to the *sml* cluster and replaced the linker and TE domain in the SmlC PKS module with the docking domain of NatC.” (Page 8)

Reviewer #3

1. The authors have altered the manuscript to make it clearer that apart from the starter module and the first module of the NRPS, the other modules are more distantly related. However in the Figure 1 legend the authors are still claiming that module 2 is conserved throughout the three systems whereas in reality the three modules incorporate different units (pyruvate, isoleucic acid, valic acid) and SmlB_A2 only shows 34% similarity to AntC_A2 of, while in fact SmlB_A3 which the authors claim is a relative insertion shows higher similarity (61%) to AntC_A2 suggesting if any module has been inserted that module 2 is the more likely candidate. The insertion of module 3 in neoantimycin relative to antimycin (NatB) is not discussed at all in the figure 1 legend.

Thank you very much for the thoughtful comments. We revised the Fig. 1 legend of as follows.

“The module and domain organizations of the starter, first, and second modules are identical among the all three systems. The starter and the first module respectively uptake 3-FSA and L-threonine in common, but the second modules uptake pyruvate (AntC), isoleucic acid (SmlB), and valic acid (NatB), respectively. JBIR-06 and neoantimycin systems include an additional NRPS modules (module 3) after the module 2 in the same ORF. The module 3 in SmlB uptakes phenylpyruvic acid, and the module 3 in NatB uptakes leucic acid. The PKS modules of JBIR-06 and neoantimycin systems (module 4) contain MT domain between AT and ACP, and accept malonyl-CoA as an extender unit, to yield dimethyl group at the α position of polyketide moiety, differently from antimycin system. Furthermore, neoantimycin system includes an extra NRPS module (module 5 as NatD) which uptakes isoleucic acid or valic acid.”

2. Page 6, line 136 more accurately describes the relationships between the clusters by only suggesting that only module 1 are conserved “derived from gene duplication of the same ancestor”.

The figure legend and the text should be brought into agreement with each other. Additionally Supplementary figure 4, which displays the percentage similarity between AntC, SmlB and NatB is hard to read. The modules with highest similarity and therefore predicted to have an ancestor in common (e.g. Greater than 79% similarity in this figure) should be highlighted to make these related modules more obvious and the modules that are closely related (maybe greater than 61% in this figure) highlighted in a different shading/colour. Since the authors are suggesting that they are following an evolutionary approach to hybrid NRPS/PKS engineering then these evolutionary relationships/differences should be clearer.

We modified Figure S4 as the reviewer suggested. Please also see our response above (Reviewer #2-3).

3. The authors have now detailed what the differences are between the NatD they sequenced for this study and the example found in the literature and explain that the previous annotation is not available for direct comparison (Page 5 line 123). However it seems that there are more variations between the two clusters other than NatD. Comparison with the previously published annotation reveals that NatE also varies between the two studies - the identically named genes are located in opposite reading frames and have dissimilar functions (MbtH in this study and PKS in the previous). The direction and size of NatG-NatR are also in disagreement with different assigned functions (based on BLAST analysis) in each study. This potentially suggests that the two sequences may originate from different organisms and/or slightly different clusters, or may just arise from differences in sequencing quality between the two studies. Without having access to the data from the other study it is impossible to say why these disparities occur but although these do not ultimately change the outcome of the work they should not be hidden and perhaps a sentence detailing the exact differences should be included for completions sake (or a supplementary figure).

According to the suggestion, we modified the Figure S2 legend. Please also see our response above (Reviewer #1-1).

4. Page 8 line 186 – The authors attribute the lack of KR (NatF) activity with the ring contracted neoantimycin to polar effects on expression of NatF. What is the evidence for this? Perhaps the enzyme no longer recognises the substrate? What is the sequence similarity to AntM? Do these discrete KR domains often show relaxed substrate specificity?

Thank you for the thoughtful comment. We agree that it is also possible to reason that NatF no longer recognizes the substrate. We revised the text as follows. "It might be caused by the polar effect on the expression of *natF*, a ketoreductase gene located downstream of *natD*, though we cannot eliminate the possibility that NatF no longer recognizes the substrate." (Page 8) However, currently, we do not have any experimental data on the substrate specificity for NatF. We fully expect to complete these studies but we feel that the present result can stand alone in terms of substance and novelty as a communication of an important discovery.

5. The term ACP is used in figure 1 and the text while T domain is used in figures 3, 4 and 5. This should ideally be consistent throughout the manuscript.

According to the suggestion, we revised the text.

6. *The discussion section contains a lot of repetition from the results section. I would recommend limiting the discussion to some simple concluding remarks and outlooks such as those starting on page 12 line 292.*

Thank you very much for the comment. According to the suggestion, we deleted some repetitive sentences, however, we would like to keep most of the part as the present form. We believe that the detailed discussion is required to increase the importance of this study.

7. *I suggest changing the first sentence in the introduction to 'are an important class of natural products' rather than major resources as written.*

Done.

8. *Page 3 line 59 – The sentence 'which defines the order of chain transfer to avoid unwanted transfer' should be reworded so the meaning is clearer.*

Done.

9. *Page 3 line 65 refers to 'these engineering studies' but no engineering studies seem to have been cited prior to that claim.*

We revised the text as follows. "the engineering of these systems usually accompanies significant loss of productivity due to the strict regulation of the module enzymes¹⁰." (page 3)

10. *Page 3 line 69 should be changed to something like 'using exchange units that are sets of A-T-C domains that can be transplanted into active chimeric modules'.*

According to the suggestion, we revised the text as follows. "using exchange units that are sets of A-T-C domains that can be transplanted into active chimeric modules." (page 3)

11. *Page 4 line 83 change 'with' to 'while'*

Done.

12. *Page 4 line 86 consider rewording to 'could lead to computational platforms which clearly predict...'*

According to the suggestion, we revised the text as follows. "could lead to computational platforms which clearly predict domain/linker/docking-domain boundaries as the one established in *cis*-PKS system for designing chimeric modules¹⁷."

13. *On Page 6, to improve the flow of the text, the paragraph beginning line 150... 'As a result' to Page 7 line 157 'NRPS-PKS system' which details the bioinformatic annotation should be moved up to line 142 and go before the sentence starting 'therefore with the bioinformatics information'.*

Done.

14. *Page 7 – line 170 – 175. This paragraph should be reworded so that its meaning is clearer.*

Done. Please also see our responses above (Reviewer #1-2).

15. Page 8 line 185 – consider changing ‘reduction’ to ‘loss’ as reduction is used in its chemical sense elsewhere in the paper.

Done.

Reviewers' comments:

Reviewer #2 (Remarks to the Author):

Begging the authors' patience, I have two further comments to make on the manuscript, prior to acceptance:

1. In the previous round, I wrote:

'...they should precisely state which module/subunit of each system corresponds to the other, as this formed the basis for their sequence comparisons (in this context, an additional figure in the supplementary section would be quite useful).'

While they have indeed provided an additional figure (Fig. S4b), it doesn't make clear the proposed evolutionary relationships between the modules in the three systems. The authors in fact now state that only module 1 has a shared origin, which would seem to undermine their argument that their engineering was based on strong evolutionary relationships between the multienzymes which allowed for comparative sequence analysis. It remains possible, however, they are missing something by comparing only the A domains (which might diverge based on their different substrate specificities), while the rest of the modules might be more similar (i.e. a pathway might be envisioned in which modules were duplicated or transferred intact between systems, and then A domains exchanged, explaining their observed overall low homology).

On the other hand, they might in fact not want to make too much of their analysis of the NRPS portions of the systems, because their engineering was focused on the PKS module – and here at least, the evolutionary relationship seems much clearer!

Overall, it would seem that more reflection is warranted on this point, as well as suitable revision of all relevant portions of the ms and figures.

2. The analysis presented in Fig. S6 is misleading, because in fact that the authors have combined C- and N-terminal sequences containing putative docking domains from both cis-AT and trans-AT PKS hybrids with NRPS (i.e. OzmK, KirAII), when the types of docking domains known to date from cis-AT and trans-AT PKS are different. Therefore, they should only include such terminal sequences from cis-AT PKS/NRPS hybrids, as with the three systems investigated here. More examples can be found in ref. 9, and the alignment adjusted to show the cluster of negatively-charged residues at the extreme end of the C-terminal docking domains that is important (the domains are quite variable, and so if upstream more conserved regions (such as carrier proteins) are included, this will bias the alignment towards these sequences). In addition, the subunit abbreviations should be defined in the legend.

3. Also for this sentence:

'The amino acid sequences within subunits is referred to as a linker domain, and those which are located on distinct subunits and mediate communication between modules are referred to as a docking domain'

Suggest:

'The amino acid sequences within subunits that join domains and modules covalently are referred to as 'linkers'...

Reviewer #3 (Remarks to the Author):

The authors have addressed the comments. The relationships between the three clusters are now much better highlighted and the similarities between each module easier to follow. The removal of some of the repetition in the discussion is welcome and it does read better for publication.

Reviewer #2

1. *'...they should precisely state which module/subunit of each system corresponds to the other, as this formed the basis for their sequence comparisons (in this context, an additional figure in the supplementary section would be quite useful).'*

While they have indeed provided an additional figure (Fig. S4b), it doesn't make clear the proposed evolutionary relationships between the modules in the three systems. The authors in fact now state that only module 1 has a shared origin, which would seem to undermine their argument that their engineering was based on strong evolutionary relationships between the multienzymes which allowed for comparative sequence analysis. On the other hand, perhaps they are missing something by comparing only the A domains (which might diverge based on their different substrate specificities), while the rest of the modules might be more similar (i.e. a pathway might be envisioned in which modules were duplicated or transferred intact between systems, and then A domains exchanged, explaining their observed overall low homology). These points need to be clarified in the text and the accompanying figures.

We appreciate the thoughtful suggestion again. We indeed learned a lot from this reviewer. According to the suggestion, we newly conducted bioinformatic analyses for C, T, and KR domains in AntC, SmlB, NatB, and NatD, respectively. We described the results in the updated Figure. S4. These points were clarified in the text.

*"We conducted the DNA sequence alignment for A, C, T, and KR domains from *ant*, *sml*, and *nat* system respectively, and hypothesized the evolutionary relationship among them based on the identity as below (Supplementary Fig. 4). The SmlB module 1 is evolved from AntC module 1 via duplication, SmlB module 2 is from the module from other organism, and SmlB module 3 is from SmlB module 2 via duplication. The modules 1-3 NatB are evolved from those of SmlB, and the module 5 NatD is likely to be derived from the module 2 SmlB, based on the high identities of their T and KR domains (76.2 and 67.9%)."* (page 6)

2. *The analysis presented in Fig. S6 is misleading, because in fact that the authors have combined C- and N-terminal sequences containing putative docking domains from both cis-AT and trans-AT PKS hybrids with NRPS (i.e. OzmK, KirAll), when the types of docking domains known to date from cis-AT and trans-AT PKS are different. Therefore, they should only include such terminal sequences from cis-AT PKS/NRPS hybrids, as with the three systems investigated here. More examples can be found in ref. 9, and the alignment adjusted to show the cluster of negatively-charged residues at the extreme end of the C-terminal docking domains that is important (the domains are quite variable, and so if upstream more conserved regions (such as carrier proteins) are included, this will bias the alignment towards these sequences). In addition, the subunit abbreviations should be defined in the legend.*

As the reviewer suggested, we removed the N- and C-terminal docking domain sequences of OzmK and KirAll from the alignments. As a result, one conserved negatively-charged residue clearly appeared in the alignment of C-terminal sequences (please see the revised Figure S6). We found the sequence of docking domains from another PKS-NRPS system in Ref #9, but did not pick them up for alignment, because all of them are the intermodule docking domains in the single protein, while the sequences that we picked are the docking domains from the two distinct proteins. We added the subunit abbreviations as the reviewer suggested.

3. *Also for this sentence:*

'The amino acid sequences within subunits is referred to as a linker domain, and those which are located on distinct subunits and mediate communication between modules are referred to as a docking domain'

Suggest:

'The amino acid sequences within subunits that join domains and modules covalently are referred to as 'linkers'...

Thank you for the suggestion. We revised the text as follows.

"The amino acid sequences within subunits that join domains and modules covalently are referred to as a linker" (Page 3)

Reviewer #3

1. The authors have addressed the comments. The relationships between the three clusters are now much better highlighted and the similarities between each module easier to follow. The removal of some of the repetition in the discussion is welcome and it does read better for publication.

Thank you very much for the suggestion. We revised the text in the discussion as follows.

“In this study, we accomplished three manipulations of the antimycin-type NRPS-PKS assembly lines: ~~ring contraction, ring expansion, and alkyl chain diversification~~, and obtained 9 novel depsipeptides (**4**, **5**, **6**, **7a-f**) with different lactone ring sizes in substantial yields. ~~Considering that there has been very little successful experiments where the size of the chemical scaffold has been rationally engineered~~, It is quite remarkable that the yields of the compounds which we obtained in the engineered NRPS-PKS system are 5-10 times higher than those in the reported module assembly line engineering^{11,48-50}. ~~This accomplishment was done with guidance of bioinformatic analysis of the co-evolved module structures~~. For the ring reduction approach, the linker between the ACP and TE domains smoothly connected the TE_{SmlC} domain to the ACP_{NatC}, resulting in the production of the tri-lactone **4** without significant drop of yield, ~~compared with **3a** (**4**: 3.9 mg/L, **3a**: 12 mg/L, Table 1)~~.” (Page 10)

“In our alkyl chain diversification approach, we introduced the broad substrate specificity of AntD into SmlC ~~by mutating the AT substrate definition sequence~~.” (Page 11)

~~“The utility of CCR enzymes¹⁹⁻²² for PKS engineering has also been demonstrated in this approach, as AntEV350G synthesized butyl-, 3-methylbutyl, and hexylmalonyl-CoAs and they were accepted by the mutated AT in SmlC. In the case of the systems with relaxed substrate specificity for acyl-CoAs, the CCR enzyme family¹⁹⁻²² could be used to diversify product structures in future.”~~ (Page 11)

“Through sequence comparisons, we can learn how nature uses “cut and paste” of module structures, which leads to the creation of novel molecules ~~by emulating nature's way~~.” (Page 12)

We hope you will agree that the manuscript has been significantly improved, and that you will find it acceptable for publication.

REVIEWERS' COMMENTS:

Reviewer #2 (Remarks to the Author):

Prior to acceptance of the manuscript, I am afraid that I have several additional comments to address.

1. Page 4, line 74: the evidence for NRPSs being monomers is not only based on chromatography, but a large number of high-resolution structures obtained by X-ray crystallography (see the recent review by Schmeing, et al. (doi: 10.1016/j.sbi.2018.01.011)).

2. Page 6, paragraph from 137-145 : If two modules from distinct systems share homology, what this means is that they are LIKELY to be derived from a common ancestral module. It is not possible, as the authors have done, to say which module evolved from which (and 'duplication' is not the right word I believe, because this rather occurs within one organism, not between organisms (horizontal transfer)). Also what, for example, are the cut-offs to use in terms of sequence identity to decide whether two modules evolved from the same ancestor (they have used 65% identity (thus excluding an example at 63% identity), but why?)

Overall, as the engineering reported in this paper targeted the PKS subunits of the systems (NatC and SmlC), while the docking domain experiment (creating an interface between SmlC and NatD) didn't rely in any sense on understanding the evolution of the NRPS portions, I would suggest completely removing all discussion of the evolution of the NRPS regions of the systems. The focus could instead be on the fact that the PKS subunits present in each system show enough mutual sequence similarity to allow for the clear identification of domain/linker boundaries (as shown in Fig. 2 and Fig. S5).

If the authors would like to maintain their evolutionary analysis, they must make it considerably more rigorous via expert advice from those in the field working on PKS and NRPS evolution (e.g. Tilmann Weber, Marnix Medema, Michael Fischbach, Jorn Piel, etc.).

3. Concerning the docking, I'm afraid that I haven't properly done my job as a reviewer (and expert), because I should have suggested from the outset that they look for additional docking domains at the remaining junctions in the Nat and other systems investigated in the paper, in order to improve the alignment in Fig. S6. An alignment based on six sequences, and which only serves to show that the putative docking domains of NatC and NatD are different from the others, isn't particularly informative. Furthermore, the legend to the figure isn't correct, as the interaction between TubB CDD and TubC NDD occurs at an NRPS/NRPS and not a PKS/NRPS interface. Also, 'coelibactin' should read 'colibactin'.

Page 11, line 262: again, replace 'intermodule linker' with 'intersubunit docking domain'

Also the following sentence (page 11, 265-267) should be modified: 'However, the interactions between SmlC and NatD are likely to be different from the known systems, because the N-terminus of NatD lacks a predicted secondary structure consistent with the crystal structure of the docking domain from the EpoB NRPS.'

It's not that the interactions between SmlC and NatD are different, because indeed this is a non-native (engineered) interaction, but that the natural interface between NatC and NatD is likely to be different than those which have been characterized to date from hybrid PKS/NRPS. Of course the NatD N-terminus cannot have the same structure as that of EpoB, because the sequence lengths are totally different (8 vs. 59 residues!).

(I would also note in passing that the response to the referee is not correct. It is not true that

docking domain alignment from ref. 9 (that presented in the main text, which shows a set of N-terminal DDs) includes docking domains from within a single protein, as by definition, docking domains act at interprotein junctions. Selected C-terminal partners of these domains are presented in the supplementary section, where it is clear that there is a patch of negatively-charged residues at the extreme C-terminus (at least two and sometimes three.)

4. Page 8, line 182: Red/ET not Rec/ET

5. Page 10, line 246 (5 +/- 2, not 5.0 +/- 1.7)

RE: NCOMMS-18-07714-T
"Reprogramming of the antimycin NRPS-PKS assembly lines inspired by gene evolution"

Reviewer #2

1. *Page 4, line 74: the evidence for NRPSs being monomers is not only based on chromatography, but a large number of high-resolution structures obtained by X-ray crystallography (see the recent review by Schmeing, et al. (doi: 10.1016/j.sbi.2018.01.011)).*
We added the new reference (ref.13) as the reviewer suggested.

2. *Page 6, paragraph from 137-145 : If two modules from distinct systems share homology, what this means is that they are LIKELY to be derived from a common ancestral module. It is not possible, as the authors have done, to say which module evolved from which (and 'duplication' is not the right word I believe, because this rather occurs within one organism, not between organisms (horizontal transfer)). Also what, for example, are the cut-offs to use in terms of sequence identity to decide whether two modules evolved from the same ancestor (they have used 65% identity (thus excluding an example at 63% identity), but why?)*

Overall, as the engineering reported in this paper targeted the PKS subunits of the systems (NatC and SmlC), while the docking domain experiment (creating an interface between SmlC and NatD) didn't rely in any sense on understanding the evolution of the NRPS portions, I would suggest completely removing all discussion of the evolution of the NRPS regions of the systems. The focus could instead be on the fact that the PKS subunits present in each system show enough mutual sequence similarity to allow for the clear identification of domain/linker boundaries (as shown in Fig. 2 and Fig. S5).

We agree with the reviewer's point that we do not rely on the evolutionary way of NRPS modules. Thus, we removed the discussion regarding the evolutionary analysis of NRPS regions as suggested, and modified the sentence accordingly.

"We conducted the DNA sequence alignment for A, C, T, and KR domains from *ant*, *sml*, and *nat* system respectively and calculated their identities (Supplementary Fig. 4)." (Page 6)

3. *Concerning the docking, I'm afraid that I haven't properly done my job as a reviewer (and expert), because I should have suggested from the outset that they look for additional docking domains at the remaining junctions in the Nat and other systems investigated in the paper, in order to improve the alignment in Fig. S6. An alignment based on six sequences, and which only serves to show that the putative docking domains of NatC and NatD are different from the others, isn't particularly informative.*

(I would also note in passing that the response to the referee is not correct. It is not true that docking domain alignment from ref. 9 (that presented in the main text, which shows a set of N-terminal DDs) includes docking domains from within a single protein, as by definition, docking domains act at interprotein junctions. Selected C-terminal partners of these domains are presented in the supplementary section, where it is clear that there is a patch of negatively-charged residues at the extreme C-terminus (at least two and sometimes three).)

Thank you very much for the thoughtful suggestion. We looked into the reference, Richter, C. D., Nietlispach, D., Broadhurst, R. W. & Weissman, K. J. Multienzyme docking in hybrid megasynthetases. *Nat. Chem. Biol.* **4**, 75–81 (2008), and picked up the additional two sets of C-terminal and N-terminal docking domains. We conducted the alignment with these two sets of sequences, and updated Figure S6. However, we did not add the docking sequence which has high similarity to NatC or D module, because we cannot find any candidate sequence in Genbank. We believe that it is sufficient to present the information that the putative docking domains in NatC and NatD are different from the others.

Furthermore, the legend to the figure isn't correct, as the interaction between TubB CDD and TubC NDD occurs at an NRPS/NRPS and not a PKS/NRPS interface. Also, 'coelibactin' should read 'colibactin'.

We do not describe the TubBC docking domain in the original version of figure legend. Nonetheless, we modified the figure legend to correct some mistypo including "coelibactin".

Page 11, line 262: again, replace 'intermodule linker' with 'intersubunit docking domain'

We corrected it as suggested by the reviewer.

Also the following sentence (page 11, 265-267) should be modified: 'However, the interactions between SmIC and NatD are likely to be different from the known systems, because the N-terminus of NatD lacks a predicted secondary structure consistent with the crystal structure of the docking domain from the EpoB NRPS.'

It's not that the interactions between SmIC and NatD are different, because indeed this is a non-native (engineered) interaction, but that the natural interface between NatC and NatD is likely to be different than those which have been characterized to date from hybrid PKS/NRPS. Of course the NatD N-terminus cannot have the same structure as that of EpoB, because the sequence lengths are totally different (8 vs. 59 residues!).

Thank you very much for the suggestion. We corrected it as below.

"The interactions between NatC and NatD are likely to be different from the known systems, because the N-terminus of NatD lacks a predicted secondary structure consistent with the crystal structure of the docking domain from the EpoB NRPS⁹." (Page10)

4. Page 8, line 182: Red/ET not Rec/ET

We corrected it.

5. Page 10, line 246 (5 +/- 2, not 5.0 +/- 1.7)

We did not correct it, because we want to keep the double-digits style for the yields.

Other

We updated Figure 1 to correct the structure of alkyl group of antimycin (1).

We hope you will agree that the manuscript has been significantly improved, and that you will find it acceptable for publication.